# Error Feedback under $(L_0, L_1)$-Smoothness: Normalization and Momentum

**Sarit Khirirat**
KAUST[*]
sarit.khirirat@kaust.edu.sa

**Abdurakhmon Sadiev**
KAUST
abdurakhmon.sadiev@kaust.edu.sa

**Artem Riabinin**
KAUST
artem.riabinin@kaust.edu.sa

**Eduard Gorbunov**
MBZUAI[†]
eduard.gorbunov@mbzuai.ac.ae

**Peter Richtárik**
KAUST
peter.richtarik@kaust.edu.sa

## Abstract

We provide the first proof of convergence for normalized error feedback algorithms across a wide range of machine learning problems. Despite their popularity and efficiency in training deep neural networks, traditional analyses of error feedback algorithms rely on the smoothness assumption that does not capture the properties of objective functions in these problems. Rather, these problems have recently been shown to satisfy generalized smoothness assumptions, and the theoretical understanding of error feedback algorithms under these assumptions remains largely unexplored. Moreover, to the best of our knowledge, all existing analyses under generalized smoothness either i) focus on single-node settings or ii) make unrealistically strong assumptions for distributed settings, such as requiring data heterogeneity, and almost surely bounded stochastic gradient noise variance. In this paper, we propose distributed error feedback algorithms that utilize normalization to achieve the $\mathcal{O}(1/\sqrt{K})$ convergence rate for nonconvex problems under generalized smoothness. Our analyses apply for distributed settings without data heterogeneity conditions, and enable stepsize tuning that is independent of problem parameters. Additionally, we provide strong convergence guarantees of normalized error feedback algorithms for stochastic settings. Finally, we show that due to their larger allowable stepsizes, our new normalized error feedback algorithms outperform their non-normalized counterparts on various tasks, including the minimization of polynomial functions, logistic regression, and ResNet-20 training.

## 1 Introduction

Machine learning models achieve impressive prediction and classification power by employing sophisticated architectures, comprising vast numbers of model parameters, and requiring training on massive datasets. Distributed training has emerged as an important approach, where multiple

---

[*]Sarit Khirirat, Abdurakhmon Sadiev, Artem Riabinin, and Peter Richtárik are with the Center of Excellence for Generative AI, King Abdullah University of Science and Technology (KAUST), Thuwal, Saudi Arabia.

[†]Eduard Gorbunov is with the Department of Statistics and Data Science, Mohamed bin Zayed University of Artificial Intelligence (MBZUAI), Abu Dhabi, United Arab Emirates.

39th Conference on Neural Information Processing Systems (NeurIPS 2025).

machines with their own local training data collaborate to train a model efficiently within a reasonable time. Many optimization algorithms can be easily adapted for distributed training frameworks. For example, stochastic gradient descent (SGD) can be modified into distributed stochastic gradient descent within a data parallelism framework, and into federated averaging algorithms [1] in a federated learning framework. However, the communication overhead of running these distributed algorithms poses a significant barrier to scaling up to large models. For example, training the VGG-16 model [2] using distributed stochastic gradient descent involves communicating $138.34$ million parameters, thus consuming over 500MB of storage and posing an unmanageable burden on the communication network between machines.

One approach to mitigate the communication burden is to apply compression. In this approach, the information, such as gradients or model parameters, is compressed using sparsifiers or quantizers to be transmitted with much lower communicated bits between machines. However, while this reduces communication overhead, too coarse compression often brings substantial challenges in maintaining high training performance due to information loss, and in extreme cases, it may potentially lead to divergence. Therefore, error feedback mechanisms have been developed to improve the convergence performance of compression algorithms, while ensuring high communication efficiency. Examples of error feedback mechanisms include EF14 [3, 4, 5, 6, 7], EF21 [8, 9], EF21-SGDM [10], EF21-P [11], and EControl [12]. Several studies developing error feedback algorithms often assume the smoothness of an objective function, i.e., its gradient is Lipschitz continuous.

However, many modern learning problems, such as distributionally robust optimization [13] and deep neural network training, are often non-smooth. For instance, the gradient of the loss computed for deep neural networks, such as LSTM [14], ResNet20 [14], and transformer models [15], is not Lipschitz continuous. These empirical findings highlight the need for a new smoothness assumption. One such assumption is $(L_0, L_1)$-smoothness, originally introduced by Zhang et al. [14], for twice differentiable functions, and later extended to differentiable functions by Chen et al. [16].

To solve generalized smooth problems, clipping and normalization have been widely utilized in first-order algorithms. Gradient descent with gradient clipping was initially shown by Zhang et al. [14] to achieve lower iteration complexity, i.e., fewer iterations needed to attain a target solution accuracy, than classical gradient descent. Subsequent works have further refined the convergence theory of clipped gradient descent [17], and improved its convergence performance by employing momentum updates [18], variance reduction techniques [19], and adaptive step sizes [20, 21, 22]. Similar convergence results have been obtained for gradient descent using normalization [23], and its momentum variants [24], including generalized SignSGD [15]. However, these first-order algorithms have mostly been explored in training on a single machine. To the best of our knowledge, distributed algorithms under generalized smoothness have been investigated in only a few works, e.g., by Crawshaw et al. [25], Liu et al. [26]. Nonetheless, these works rely on assumptions limiting families of optimization problems, including data heterogeneity, almost sure variance bounds, and symmetric noise distributions around the mean assumptions. Furthermore, these first-order algorithms under generalized smoothness do not incorporate compression techniques to improve communication efficiency. These aspects motivate us to develop *distributed communication-efficient algorithms for solving nonconvex generalized smooth problems.*

## 1.1 Contributions

In this paper, we develop distributed error feedback algorithms for communication-efficient optimization under nonconvex, generalized smooth regimes. Our contributions are summarized below.

● **Importance of normalization.** Just as gradient clipping is crucial for gradient descent, we empirically demonstrate that normalization stabilizes the convergence of error feedback algorithms for minimizing nonconvex generalized smooth functions. In this paper, we introduce a variant of EF21, a widely used error feedback algorithm by Richtárik et al. [8], which incorporates normalization to guarantee convergence for nonconvex, generalized smooth problems. In a single-node setting, this new method, which we call ||EF21-GD||, or more compactly as ||EF21||, provides larger stepsize, and faster convergence rate than its non-normalized counterpart EF21 for minimizing simple nonconvex polynomial functions that satisfy generalized smoothness, as shown by Figure 1.

● **Convergence of normalized error feedback algorithms.** We establish an $\mathcal{O}(1/\sqrt{K})$ convergence rate in the gradient norm for ||EF21|| on nonconvex generalized smooth problems. ||EF21|| achieves

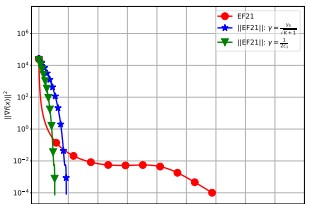 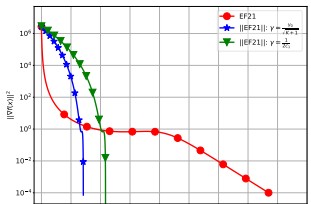 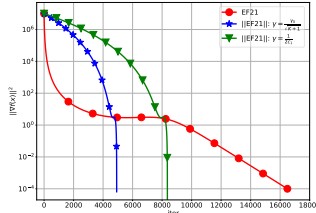

Figure 1: The minimization of polynomial functions using EF21 with $\gamma = \frac{1}{L + L\sqrt{\frac{\beta}{\theta}}}$, and ||EF21|| with $\gamma = \frac{\hat{\gamma}}{\sqrt{K+1}}$, $\hat{\gamma} = 1$ (blue line) and $\gamma = \frac{1}{2c_1}$ (green line). Here, we ran both algorithms for (1) $L_0 = 4$, $L_1 = 1$, and $K = 2,000$ (left), (2) $L_0 = 4$, $L_1 = 4$, and $K = 5,000$ (middle), and (3) $L_0 = 4$, $L_1 = 8$, and $K = 16,000$ (right).

the same rate as EF21 under $L$-smoothness by [8]. Our results are derived under standard assumptions, i.e., generalized smoothness and the existence of lower bounds on the objective function, and are applicable in distributed settings regardless of any data heterogeneity degree, unlike the results by Crawshaw et al. [25], Liu et al. [26]. Additionally, our stepsize rules for ||EF21|| ensure convergence without requiring knowledge of the generalized smoothness constants $L_0$ or $L_1$, in contrast to Richtárik et al. [8], where the stepsize depends on the smoothness constant $L$ (which is often inaccessible).

• **Extension to stochastic settings.** Furthermore, we propose a variant of EF21-SGDM, an error feedback algorithm with momentum updates by Fatkhullin et al. [10], that employs normalization for solving nonconvex, stochastic optimization under generalized smoothness. Specifically, we prove that ||EF21-SGDM|| with suitable stepsize choices attains the same $\mathcal{O}(1/K^{1/4})$ convergence rate in the gradient norm as EF21-SGDM.

• **Numerical evaluation.** We implemented ||EF21|| using the stepsize rules derived from our theory, and compared its performance against EF21. Both algorithms were evaluated on three learning tasks: minimizing nonconvex polynomial functions, solving logistic regression with a nonconvex regularizer, and training ResNet-20 on the CIFAR-10 dataset. Thanks to its larger stepsizes, ||EF21|| outperforms EF21, in terms of both convergence speed and solution accuracy across these tasks.

| Methods | Complexity | Smoothness | Variance bound | Normalization |
|---|---|---|---|---|
| EF21
Richtárik et al. [8] | $\mathcal{O}(1/\epsilon^2)$ | $L$ | No | No |
| EF21-SGDM
Fatkhullin et al. [10] | $\mathcal{O}(1/\epsilon^4)$ | $L$ | expectation | No |
| ||EF21||
**NEW** (Alg. 1) | $\mathcal{O}(1/\epsilon^2)$ | $(L_0, L_1)$ | No | Yes |
| ||EF21-SGDM||
**NEW** (Alg. 2) | $\mathcal{O}(1/\epsilon^4)$ | $(L_0, L_1)$ | Expectation | Yes |

Table 1: Comparisons of complexities and assumptions between known and our results for EF21 variants. The complexity is defined by the iteration count $K$ required by the algorithms to attain $\min_{k=0,1,\ldots,K} \mathrm{E}\left[\left\|\nabla f(x^k)\right\|\right] \leq \epsilon$. $(L_0, L_1)$-smoothness refers to generalized smoothness in Assumption 3. The variance bound in expectation is defined in Assumption 5.

## 2 Related Works

**Error feedback.** Error feedback mechanisms have been utilized in various algorithms with communication compression, leading to significant improvements in solution accuracy, while reducing communication. As the first version of these mechanisms, EF14 was introduced by Seide et al. [3], and later analyzed for first-order algorithms in both single-node [4, 27] and distributed settings [5, 6, 28, 29, 7, 30, 31, 32]. Next, EF21 is another error feedback variant proposed by Richtárik et al. [8], which offers strong convergence guarantees for distributed gradient algorithms with any

contractive compressors, without requiring bounded gradient norm or bounded data heterogeneity assumptions. EF21 can also be adapted for stochastic optimization through sufficiently large mini-batches [9] or momentum updates [10]. More recently, EControl was developed by Gao et al. [12] to guarantee provably superior complexity results for distributed stochastic optimization compared to prior error feedback mechanisms. To the best of our knowledge, these existing works on error feedback have focused solely on optimization under traditional $L$-smoothness. In this paper, we introduce a normalized variant of the EF21 methods [8] for solving nonconvex generalized smooth problems. In particular, we prove that ||EF21|| under generalized smoothness achieves the same $\mathcal{O}(1/\sqrt{K})$ rate as EF21 under traditional smoothness, and demonstrate in experiments that ||EF21|| permits larger step sizes, and thus attains faster convergence than EF21.

**Non-smoothness assumptions.** Empirical findings suggest that the traditional smoothness used for analyzing optimization algorithms does not capture the properties of objective functions in many machine learning problems, especially deep neural network training problems. This motivates researchers to consider different assumptions to replace this traditional smoothness condition. First introduced by Zhang et al. [14], the $(L_0, L_1)$-smoothness condition on a twice differentiable function $f(x)$ is defined by $\left\| \nabla^2 f(x) \right\| \le L_0 + L_1 \left\| \nabla f(x) \right\|$ for $x \in \mathbb{R}^d$. This $(L_0, L_1)$-smoothness has been extended to differentiable functions without assuming the existence of the Hessian. For instance, the smoothness with a differentiable function $\ell(x)$ [33], and symmetric generalized smoothness [16] cover the $(L_0, L_1)$-smoothness when the Hessian exists, and includes many important machine learning problems, such as phase retrieval problems [16], and distributionally robust optimization [34]. Other classes of non-smoothness assumptions, which are not related to the generalized smoothness but capture other optimization problems, include Hölder's continuity of the gradient [35], the relative smoothness [36], and the polynomial growth of the gradient norm [37]. In this paper, we impose the generalized smoothness condition to establish the convergence of ||EF21|| for solving deterministic and stochastic optimization.

**Gradient clipping and normalization.** Clipping and normalization are commonly employed in gradient-based methods for solving generalized smooth problems. Clipped (stochastic) gradient descent has been studied for both nonconvex and convex problems under $(L_0, L_1)$-smoothness conditions by Zhang et al. [14], Koloskova et al. [17]. Extensions to clipped gradient algorithms have been proposed, including momentum updates [18], variance reduction methods [19], and adaptive step sizes [20, 21, 22, 38]. Comparable complexities have been achieved for normalized gradient descent [23], and its momentum-based variants [24], including SignSGD [15] and its variance-reduction variants [39]. Convergence properties of gradient-based algorithms have also been explored under more generalized forms of non-uniform smoothness, extending beyond the $(L_0, L_1)$-smoothness by Zhang et al. [14] to cover a wider range of optimization problems. For example, variants of (stochastic) gradient descent have been analyzed under $\alpha$-symmetric generalized smoothness by Chen et al. [16], and under $\ell$-smoothness involving certain differentiable functions $\ell(\cdot)$ by Li et al. [33, 21]. However, the majority of these analyses focus on the single-node setting. To the best of our knowledge, only a limited number of works, such as those by Crawshaw et al. [25], Liu et al. [26], have examined federated averaging algorithms for nonconvex problems under generalized smoothness. These works, however, often rely on restrictive assumptions, including data heterogeneity, almost sure variance bounds, and symmetric noise distributions centered around their means. In this paper, we develop distributed error feedback algorithms, which eliminate the need for the restrictive assumptions mentioned above, and rely on standard assumptions on objective functions and compressors.

## 3  Preliminaries

**Notations.** We use $[n]$ to denote the set $\{1, 2, \ldots, n\}$, and $\mathrm{E}[u]$ to represent the expectation of a random variable $u$. Additionally, $\|\cdot\|$ indicates the Euclidean norm for vectors or the spectral norm for matrices, and $\|\cdot\|_1$ is the $\ell_1$-norm for vectors, while $\langle x, y \rangle$ denotes the inner product between $x$ and $y$ in $\mathbb{R}^d$. Lastly, for a square matrix $A \in \mathbb{R}^{d \times d}$, $\lambda_{\min}(A)$ refers to its minimum eigenvalue, and $I \in \mathbb{R}^{d \times d}$ is the identity matrix.

**Problem Formulation.** We focus on the following distributed optimization problem:

$$\min_{x \in \mathbb{R}^d} \left\{ f(x) := \frac{1}{n} \sum_{i=1}^{n} f_i(x) \right\}, \tag{1}$$

where $n$ refers to the number of clients, and $f_i(x)$ is the loss of a model parameterized by vector $x \in \mathbb{R}^d$ over its local data $\mathcal{D}_i$ owned by client $i \in [n]$.

**Assumptions.** To facilitate our convergence analysis, we make standard assumptions on objective functions and compression operators.

**Assumption 1** (Lower Boundedness of $f$). *The function $f$ is bounded from below, i.e.,*

$$f^{\mathrm{inf}} = \inf_{x \in \mathbb{R}^d} f(x) > -\infty.$$

**Assumption 2** (Lower Boundedness of $f_i$). *For each $i \in [n]$, the function $f_i$ is bounded from below, i.e.,*

$$f_i^{\mathrm{inf}} := \inf_{x \in \mathbb{R}^d} f_i(x) > -\infty.$$

Assumptions 1 and 2 are standard for analyzing optimization algorithms for unconstrained problems.

**Assumption 3** (Generalized Smoothness of $f_i$). *A function $f_i(x)$ is symmetrically generalized smooth if there exists $L_0, L_1 > 0$ such that for $u_\theta = \theta x + (1 - \theta)y$, and for all $x, y \in \mathbb{R}^d$,*

$$\|\nabla f_i(x) - \nabla f_i(y)\| \le \left( L_0 + L_1 \sup_{\theta \in [0,1]} \|\nabla f_i(u_\theta)\| \right) \|x - y\|. \tag{2}$$

Assumption 3 refers to symmetric generalized smoothness by Chen et al. [16], which covers asymmetric generalized smoothness [17, 16], and the original $(L_0, L_1)$-smoothness by [14]. Moreover, Assumption 3 covers the functions with unbounded classical smoothness constant, e.g., exponential function. Additionally, Assumption 3 with $L_1 = 0$ reduces to the traditional $L_0$-smoothness [40, 41], under which the convergence of optimization algorithms has been extensively studied.

**Assumption 4** (Contractive Compressor). *An operator $\mathcal{C}^k : \mathbb{R}^d \to \mathbb{R}^d$ is an $\alpha$-contractive compressor if there exists $\alpha \in (0, 1]$ such that for $k \ge 0$ and $v \in \mathbb{R}^d$,*

$$\mathrm{E}\left[ \left\| \mathcal{C}^k(v) - v \right\|^2 \right] \le (1 - \alpha) \|v\|^2. \tag{3}$$

Furthermore, compressors defined by Assumption 4 cover top-$k$ sparsifiers [5, 4], low-rank approximation [42, 43], and various other compressors described by Safaryan et al. [44], Beznosikov et al. [45], Demidovich et al. [46].

**Assumption 5** (Bounded Variance). *A stochastic gradient $\nabla f_i(x; \xi_i)$ with its sample $\xi_i \sim \mathcal{D}_i$ is an unbiased estimator of $\nabla f_i(x)$ with bounded variance, i.e., for all $x \in \mathbb{R}^d$,*

$$\mathrm{E}\left[ \nabla f_i(x; \xi_i) \right] = \nabla f_i(x), \quad \text{and} \quad \mathrm{E}\left[ \|\nabla f_i(x; \xi_i) - \nabla f_i(x)\|^2 \right] \le \sigma^2. \tag{4}$$

Assumption 5 is standard for stochastic optimization [47, 48, 49] that is only imposed on each local stochastic gradient, and it does not imply data heterogeneity, i.e., the bounded difference between each component function $f_i(x)$ and the global function $f(x)$.

## 4 Normalized Error Feedback (||EF21||)

For nonconvex deterministic optimization under generalized smoothness, we develop a distributed error feedback algorithm. One challenge is that the generalized smoothness parameter scales with the gradient norm $\|\nabla f(x^k)\|$. To resolve this issue, we apply gradient normalization to the algorithms. In particular, we consider ||EF21||, the normalized version of EF21 [8] that updates the next iterates $x^{k+1}$ using the ||EF21|| update. The full description of ||EF21|| can be found in Algorithm 1.

Our new method ||EF21||, just like EF21 [8] under traditional smoothness, enjoys the $\mathcal{O}(1/\sqrt{K})$ convergence in the gradient norm under generalized smoothness, as shown below.

**Algorithm 1** Normalized Error Feedback ($\|$EF21$\|$)

1: **Input:** Stepsize $\gamma_k > 0$ for $k = 0, 1, \ldots$; starting points $x^0, g_i^{-1} \in \mathbb{R}^d$ for $i \in \{1, 2, \ldots, n\}$; and $\alpha$-contractive compressors $\mathcal{C}^k : \mathbb{R}^d \to \mathbb{R}^d$ for $k = 0, 1, \ldots$.
2: **for** each iteration $k = 0, 1, \ldots, K$ **do**
3:     **for** each client $i = 1, 2, \ldots, n$ in parallel **do**
4:         Compute local gradient $\nabla f_i(x^k)$
5:         Transmit $\Delta_i^k = \mathcal{C}^k(\nabla f_i(x^k) - g_i^{k-1})$
6:         Update $g_i^k = g_i^{k-1} + \Delta_i^k$
7:     **end for**
8:     Central server computes $g^k = \frac{1}{n}\sum_{i=1}^n g_i^k$ via $g_i^k = g_i^{k-1} + \Delta_i^k$
9:     Central server updates $x^{k+1} = x^k - \gamma_k \frac{g^k}{\|g^k\|}$
10: **end for**
11: **Output:** $x^{K+1}$

---

**Theorem 1** (Convergence of $\|$EF21$\|$). *Consider Problem (1), where Assumption 1 (lower bound on $f$), Assumption 2 (lower bound on $f_i$), Assumption 3 (generalized smoothness of $f_i$), and Assumption 4 (contractive compressor) hold. Then, the iterates $\{x^k\}$ generated by $\|$EF21$\|$ (Algorithm 1) with*

$$\gamma_k = \frac{\gamma}{\sqrt{K+1}}$$

*for $K \geq 0$ and $\gamma > 0$ satify*

$$\min_{k=0,1,\ldots,K} \mathrm{E}\left[\left\|\nabla f(x^k)\right\|\right] \leq \frac{V^0 \exp(8c_1 L_1 \exp(L_1\gamma)\gamma^2)}{\gamma\sqrt{K+1}} + B\frac{\gamma\exp(L_1\gamma)}{\sqrt{K+1}},$$

*where $V^k := f(x^k) - f^{\inf} + \frac{2\gamma_k}{1-\sqrt{1-\alpha}}\frac{1}{n}\sum_{i=1}^n \left\|\nabla f_i(x^k) - g_i^k\right\|$, $B = 2c_0 + \frac{8L_1 c_1}{n}\sum_{i=1}^n(f^{\inf} - f_i^{\inf})$, and $c_i = \left(\frac{1}{2} + 2\frac{\sqrt{1-\alpha}}{1-\sqrt{1-\alpha}}\right)L_i$ for $i = 0, 1$.*

Theorem 1 establishes the $\mathcal{O}(1/\sqrt{K})$ convergence in the expectation of gradient norms for $\|$EF21$\|$ on nonconvex deterministic problems under generalized smoothness. This rate is the same as Theorem 1 of Richtárik et al. [8] for EF21 under traditional smoothness, and does not depend on data heterogeneity conditions in contrast to Crawshaw et al. [25], Liu et al. [26]. Also, our stepsize depends on any positive constant $\gamma_0$, and total iteration count $K$, without needing to know smoothness constants $L_0, L_1$ in contrast to Richtárik et al. [8]. Additionally, if we choose $\gamma_0 = 1/(8cL_1)$, then our convergence bound from Theorem 1 becomes

$$\min_{k=0,1,\ldots,K} \mathrm{E}\left[\left\|\nabla f(x^k)\right\|\right] \leq \frac{32cL_1 V^0 + L_0/L_1 + 2L_1\delta^{\inf}}{\sqrt{K+1}},$$

where $c = \frac{1}{2} + 2\frac{\sqrt{1-\alpha}}{1-\sqrt{1-\alpha}}$, and $\delta^{\inf} = \frac{1}{n}\sum_{i=1}^n(f^{\inf} - f_i^{\inf})$.

**Comparisons between $\|$EF21$\|$ and EF21 under traditional smoothness.** For nonconvex, traditional smooth problems, $\|$EF21$\|$ from Theorem 1 with $L_1 = 0$ achieves the same $\mathcal{O}(1/\sqrt{K})$ rate in the expectation of gradient norms as EF21 analyzed by Richtárik et al. [8], but with a larger convergence factor of $2\sqrt{2}$. We refer to the derivation and discussion in details in Appendix C.

In the following section, we demonstrate how to integrate normalization into EF21-SGDM [10], an error feedback algorithm that allows each node to compute its local stochastic gradient, for solving nonconvex stochastic problems.

## 5 Normalized Error Feedback with Stochastic Gradients & Momentum ($\|$EF21-SGDM$\|$)

Having established the convergence of $\|$EF21$\|$ for deterministic optimization, we will next develop a distributed error feedback algorithm that incorporate stochastic gradients and normalization to accommodate generalized smoothness conditions. In particular, we focus on $\|$EF21-SGDM$\|$ (Algorithm 2),

---

**Algorithm 2** Normalized Error Feedback with Stochastic Gradients & Momentum (‖EF21-SGDM‖)

---

1: **Input:** Stepsizes $\gamma_k > 0$ and $\eta_k \in [0,1]$ for $k = 0,1,\dots$; starting points $x^0, g_i^{-1} \in \mathbb{R}^d$ for $i \in \{1,2,\dots,n\}$, and $v_i^{-1} = \nabla f_i(x_i^0; \xi_i^0)$ with independent random samples $\xi_i$ for $i \in \{1,2,\dots,n\}$; $\alpha$-contractive compressors $\mathcal{C}^k : \mathbb{R}^d \to \mathbb{R}^d$ for $k = 0,1,\dots$
2: **for** each iteration $k = 0,1,\dots,K$ **do**
3:     **for** each client $i = 1,2,\dots,n$ in parallel **do**
4:         Compute a local stochastic gradient $\nabla f_i(x^k; \xi_i^k)$
5:         Update a momentum estimator $v_i^k = (1 - \eta_k)v_i^{k-1} + \eta_k \nabla f_i(x^k; \xi_i^k)$
6:         Transmit $\Delta_i^k = \mathcal{C}^k(v_i^k - g_i^{k-1})$
7:         Update $g_i^k = g_i^{k-1} + \Delta_i^k$
8:     **end for**
9:     Central server computes $g^k = \frac{1}{n}\sum_{i=1}^n g_i^k$ via $g_i^k = g_i^{k-1} + \Delta_i^k$
10:     Central server updates $x^{k+1} = x^k - \gamma_k \frac{g^k}{\|g^k\|}$
11: **end for**
12: **Output:** $x^{K+1}$

---

the normalized version of EF21-SGDM due to Fatkhullin et al. [10]. We also note that ‖EF21-SGDM‖ recovers many optimization algorithms of interest in the special cases. For instance, it reduces to

- normalized version of EF21 [8], which we call ‖EF21‖, when we let $\eta_k = 1$ and $\nabla f_i(x^k; \xi_i^k) = \nabla f_i(x^k)$,
- normalized version of EF21-SGD [9], which we call ‖EF21-SGD‖, when we let $\eta_k = 1$, and
- normalized version of SGDM [50], which we call ‖SGDM‖[3], when we let $\eta_k = 1 - \beta_k$ and $\mathcal{C}^k(\cdot)$ is the identity compressor/mapping.

In the next theorem, we demonstrate that ‖EF21-SGDM‖ attains the same $\mathcal{O}(1/K^{1/4})$ convergence rate as both EF21-SGDM and ‖SGDM‖.

**Theorem 2** (Convergence of ‖EF21-SGDM‖). *Consider Problem (1), where Assumption 1 (lower bound on $f$), Assumption 2 (lower bound on $f_i$), Assumption 3 (generalized smoothness of $f_i$), Assumption 4 (contractive compressor), and Assumption 5 (bounded variance) hold. If $g_i^{-1} = 0$ for $i \in \{1,\dots,n\}$ and*

$$\gamma_k \equiv \gamma = \frac{\gamma}{(K+1)^{3/4}}, \text{ with } 0 < \gamma \leq \frac{1}{16L_1} \min\left\{(K+1)^{1/2}C_\alpha, 1\right\}, \quad and$$

$$\eta_k \equiv \eta = \frac{1}{(K+1)^{1/2}},$$

*where $C_\alpha := 1 - \sqrt{1-\alpha}$, then the iterates $\{x^k\}$ generated by ‖EF21-SGDM‖ (Algorithm 2) satisfy for $K \geq 0$*

$$\min_{k=0,1,\dots,K} \mathrm{E}\left[\|\nabla f(x^k)\|\right] \leq \mathcal{O}\left(\frac{\delta^0/\gamma + \sigma/\sqrt{n} + \gamma(L_0 + L_1^2\delta^{\mathrm{inf}})}{(K+1)^{1/4}}\right)$$

$$+ \mathcal{O}\left(\frac{\sqrt{1-\alpha}}{\alpha}\left(\frac{\sigma}{(K+1)^{1/2}} + \frac{\gamma(L_0 + L_1^2\delta^{\mathrm{inf}})}{(K+1)^{3/4}}\right)\right),$$

*where $\delta^0 := f(x^0) - f^{\mathrm{inf}}$, and $\delta^{\mathrm{inf}} := \frac{1}{n}\sum_{i=1}^n(f^{\mathrm{inf}} - f_i^{\mathrm{inf}})$.*

From Theorem 2, ‖EF21-SGDM‖ under generalized smoothness achieves the $\mathcal{O}(1/K^{1/4})$ convergence rate in the expectation of gradient norms. This rate is the same as that of EF21-SGDM, previously analyzed under traditional smoothness by Fatkhullin et al. [10, Theorem 3]. The result holds regardless of the data heterogeneity degree and the mini-batch size. We also notice that the stepsize $\gamma_0$ for ‖EF21-SGDM‖ , unlike in the case of ‖EF21‖, depends on the generalized smoothness constant $L_1$, and the compression parameter $\alpha$. However, the considered choice of stepsizes is agnostic to $\sigma$ and $L_0$.

---

[3]This method is also known as NSGD-M.

Furthermore, Theorem 2 with $\alpha = 1$ (i.e., $\mathcal{C}^k$ is the identity compressor) implies the convergence bound of the distributed version of normalized SGD with momentum (∥SGDM∥) [50] using $\beta = 1 - \eta$:

$$\min_{k=0,1,\ldots,K} \mathrm{E}\left[\left\|\nabla f(x^k)\right\|\right] \leq \mathcal{O}\left(\frac{(f(x^0) - f^{\inf})/\gamma + \sigma/\sqrt{n} + \gamma L_0 + \gamma L_1^2 \delta^{\inf}}{(K+1)^{1/4}}\right). \tag{5}$$

For the single-node SGDM, where $n = 1$ and $\delta^{\inf} = 0$, our convergence bound in (5) with $\gamma = \Theta(1/L_1)$ achieves the $\mathcal{O}\left(\frac{L_1(f(x^0) - f^{\inf}) + \sigma + L_0/L_1}{(K+1)^{1/4}}\right)$ convergence, which matches the rate obtained by Hübler et al. [24, Corollary 3]. Unlike the earlier results for single-node SGDM, our result holds for the multi-node regime. The bound in (5) for multi-node SGDM includes the $\sigma/\sqrt{n}$-term indicating a $\sqrt{n}$-fold reduction in the influence of stochastic variance noise $\sigma$, and the $\gamma L_1^2 \delta^{\inf}$-term accounting for the effect of data heterogeneity.

**Novel proof techniques for ∥EF21∥ and ∥EF21-SGDM∥ under generalized smoothness.** Our analysis demonstrates that ∥EF21∥ achieves the convergence rate under generalized smoothness equivalent to EF21 under traditional smoothness. However, our proof techniques differ significantly from prior work. We employ different Lyapunov functions. For ∥EF21∥, we use $V^k := f(x^k) - f^{\inf} + \frac{A}{n}\sum_{i=1}^{n}\left\|\nabla f_i(x^k) - g_i^k\right\|$, in constrast to Richtárik et al. [8] that uses $V^k := f(x^k) - f^{\inf} + \frac{B}{n}\sum_{i=1}^{n}\left\|\nabla f_i(x^k) - g_i^k\right\|^2$. For ∥EF21-SGDM∥, we use $V^k := f(x^k) - f^{\inf} + \frac{C}{n}\sum_{i=1}^{n}\left\|v_i^k - g_i^k\right\| + \frac{D}{n}\sum_{i=1}^{n}\left\|v_i^k - \nabla f_i(x^k)\right\|$, unlike Fatkhullin et al. [10] that uses $V^k := f(x^k) - f^{\inf} + \frac{E}{n}\sum_{i=1}^{n}\left\|v_i^k - g_i^k\right\|^2 + \frac{F}{n}\sum_{i=1}^{n}\left\|v_i^k - \nabla f_i(x^k)\right\|^2$. These new Lyapunov functions necessitate the Lyapunov- based convergence analysis, distinct from standard techniques for error feedback methods. Our analysis leverages Lemma 2 to handle generalized smoothness. For ∥EF21∥, we rely on Lemma 4. For ∥EF21-SGDM∥, we derive a new upper-bound on $\mathrm{E}\left[\left\|v^k - \nabla f(x^k)\right\|\right]$, unlike Fatkhullin et al. [10] to show the $\sqrt{n}$-speedup for the term proportional to $\sigma$, and utilize non-uniform weights to obtain convergence in the gradient norm.

# 6 Experiments

In this section, we evaluate the performance of ∥EF21∥, and compare it against EF21 [8]. We test these algorithms for three nonconvex, generalized smooth problems: the problem of minimizing polynomial functions, the logistic regression problem with a nonconvex regularization term over synthetic and benchmark datasets from LIBSVM [51], and the training of the ResNet-20 [52] model over the CIFAR10 [53] dataset[4]. For all experiments, we use a top-$k$ sparsifier, which is a $\frac{k}{d}$-contractive compressor.

## 6.1 Logistic Regression with a Nonconvex Regularizer

First, we consider a logistic regression problem with a nonconvex regularizer, i.e., Problem (1) with

$$f_i(x) = \log(1 + \exp(-b_i a_i^T x)) + \lambda \sum_{j=1}^{d} \frac{x_j^2}{1 + x_j^2},$$

where $a_i \in \mathbb{R}^d$ is the $i^{\text{th}}$ feature vector of data matrix $A \in \mathbb{R}^{n \times d}$ with its class label $b_i \in \{-1, 1\}$, and $\lambda > 0$ is a regularization parameter. Here, $f(x)$ is nonconvex, and $L$-smooth with $L = \|A\|^2/(4n) + 2\lambda$. Also, each $f_i(x)$ is $\hat{L}_i$-smooth with $\hat{L}_i = \|a_i\|^2/4 + 2\lambda$, and generalized smooth with $L_0 = 2\lambda + \lambda\sqrt{d}\max_i\|a_i\|$ and $L_1 = \max_i\|a_i\|$. The derivations of smoothness parameters can be found in Appendix H.

In this experiment, we initialized $x^0 \in \mathbb{R}^d$, where each coordinate was drawn from a standard normal distribution $\mathcal{N}(0,1)$, and set $\lambda = 0.1$. Here, the condition $\lambda > \lambda_{\min}\left(A^\top A\right)/(2n)$ ensures that $f(x)$ is nonconvex. We ran ∥EF21∥ and EF21 on the following datasets: (1) two from LIBSVM [51]: Breast Cancer ($n = 683$, $d = 10$, and scaled to $[-1, 1]$), and a1a ($n = 1605$, $d = 123$); and (2) a synthetically generated dataset ($n = 20$, $d = 10$), where the data matrix $A \in \mathbb{R}^{n \times d}$ had entries drawn from $\mathcal{N}(0,1)$, and the class label $b_i$ was set to either $-1$ or $1$ with equal probability. For

---

[4]We implemented EF21 and ∥EF21∥ on training the ResNet-20 model by using PyTorch. Our source codes can be found in the link to error-feedback-generalized-smoothness-paper.

EF21, we selected the stepsize $\gamma_k = 1/\left(L + \tilde{L}\sqrt{\beta/\theta}\right)$ with $\tilde{L} = \sqrt{\sum_{i=1}^n \hat{L}_i^2/n}$, $\theta = 1 - \sqrt{1-\alpha}$, and $\beta = (1-\alpha)/(1-\sqrt{1-\alpha})$, given by Richtárik et al. [8, Theorem 1]. For ||EF21||, we chose $\gamma_k = \gamma/\sqrt{K+1}$ with $\gamma > 0$ from Theorem 1, by setting $\gamma_0 = 1$, $K = 100$ for the generated data and Breast Cancer, and $K = 400$ for a1a. We choose $\gamma_0 = 1$, because ||EF21|| with $\gamma_0 \in [1, 10]$ converges faster than that with small values of $\gamma_0$ (e.g. 0.1), when we run the algorithm on a single node ($n = 1$) for minimizing polynomial function and solving logistic regression. We determine $K$ as the smallest number of iterations required to achieve the desired accuracy by performing a grid search with a stepsize of 50.

Figure 2 shows that ||EF21|| outperforms the traditional EF21 on all evaluated datasets, achieving faster convergence and higher solution accuracy. This improvement results from the fact that the theoretical stepsize for ||EF21||, as derived in Theorem 1, is larger than the stepsize for EF21 outlined by Richtárik et al. [8, Theorem 1].

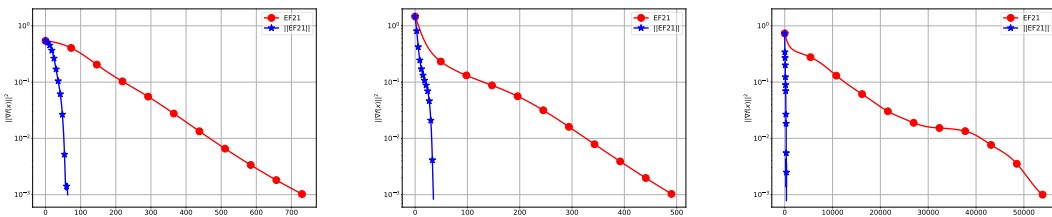

Figure 2: Logistic regression with a nonconvex regularizer using normalized ||EF21|| and EF21. We reported $\left\|\nabla f(x^k)\right\|^2$ with respect to iteration count $k$. We used the constant stepsize $\gamma = \frac{1}{L+\tilde{L}\sqrt{\frac{\beta}{\theta}}}$ for EF21, and $\gamma = \frac{\hat{\gamma}}{\sqrt{K+1}}$, $\hat{\gamma} = 1$ for ||EF21||. Here, $K = 100$ for our generated data (left), and Breast Cancer (middle), while $K = 400$ for a1a (right).

## 6.2  ResNet20 Training Over CIFAR-10

Next, we trained the ResNet20 [52] model on the CIFAR-10 [53] dataset, which was demonstrated empirically by Zhang et al. [14] to satisfy the $(L_0, L_1)$-smoothness condition. In these experiments, we used a top-$k$ compressor over 50,000 training images, with evaluation on 10,000 test images. The dataset was evenly distributed among 5 clients, each using a mini-batch size of 128. Both algorithms were run for 100 epochs with a constant stepsize $\gamma = 5$. Here, one epoch refers to a full pass through the entire dataset processed by all clients.

From Figure 3, under the same constant stepsize and the top-$k$ sparsifier with $k = 0.01d$, ||EF21|| outperforms EF21, in terms of convergence speed (in gradient norms and losses) and accuracy, relative to the number of bits communicated from each client to the server. Specifically, ||EF21|| achieved accuracy gains of up to 10% over EF21.

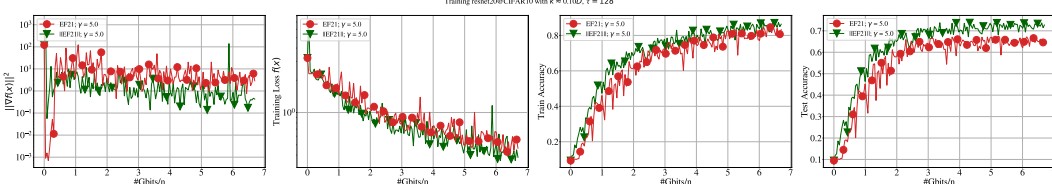

Figure 3: ResNet20 training on CIFAR-10 by using EF21 and ||EF21|| under the same stepsize $\gamma = 5$ and $k = 0.1d$ for a top-$k$ sparsifier.

## 7  Conclusion and Future Works

In this paper, we have demonstrated that normalization can be effectively combined with EF21 to develop distributed error feedback algorithms for solving nonconvex optimization problems under generalized smoothness conditions. Specifically, ||EF21|| and ||EF21-SGDM|| achieve convergence

rates of $\mathcal{O}(1/K^{1/2})$ in deterministic settings and $\mathcal{O}(1/K^{1/4})$ in stochastic settings, respectively. These convergence rates match those of the vanilla EF21 and EF21-SGDM algorithms. Unlike previous works on distributed algorithms under generalized smoothness, our analysis does not assume data heterogeneity or impose smoothness-dependent restrictions on the stepsize (in the deterministic case). Finally, our experiments confirm that ||EF21|| exhibits stronger convergence performance compared to the original EF21, due to its larger allowable stepsizes.

Our work implies many promising research directions. One interesting direction is to extend our convergence results for ||EF21|| and ||EF21-SGDM|| to accommodate decreasing or adaptive stepsize schedules, as the constant stepsizes required by our current analysis can become impractically small when the total number of iterations is large. Another important direction is the development of distributed and federated algorithms that leverage clipping or normalization for minimizing nonconvex generalized smooth functions.

### Acknowledgements

The research reported in this publication was supported by funding from King Abdullah University of Science and Technology (KAUST): i) KAUST Baseline Research Scheme, ii) CRG Grant ORFS-CRG12-2024-6460, and iii) Center of Excellence for Generative AI, under award number 5940.

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

# Contents

## A  Lemmas

In this section, we introduce useful lemmas for our analysis. Lemmas 1 and 2 introduce inequalities by generalized smoothness, while Lemmas 3 and 4 present the descent inequality and convergence rate, respectively, when the normalized gradient descent update is applied.

> **Lemma 1.** *Let each $f_i(x)$ be generalized smooth with parameters $L_0, L_1 > 0$, and lower bounded by $f_i^{\text{inf}}$, and let $f(x) = \frac{1}{n} \sum_{i=1}^{n} f_i(x)$. Then, for any $x, y \in \mathbb{R}^d$*
>
> $$\|\nabla f_i(x) - \nabla f_i(y)\| \leq (L_0 + L_1 \|\nabla f_i(y)\|) \exp(L_1 \|x - y\|) \|x - y\|, \tag{6}$$
>
> $$f_i(y) \leq f_i(x) + \langle \nabla f_i(x), y - x \rangle + \frac{L_0 + L_1 \|\nabla f_i(x)\|}{2} \exp(L_1 \|x - y\|) \|y - x\|^2, \tag{7}$$
>
> $$\frac{\|\nabla f_i(x)\|^2}{4(L_0 + L_1 \|\nabla f_i(x)\|)} \leq f_i(x) - f_i^{\text{inf}}, \text{ and} \tag{8}$$
>
> $$f(y) \leq f(x) + \langle \nabla f(x), y - x \rangle + \frac{L_0 + \frac{L_1}{n} \sum_{i=1}^{n} \|\nabla f_i(x)\|}{2} \exp(L_1 \|x - y\|) \|y - x\|^2 \tag{9}$$

*Proof.* The first and second statements are derived in Chen et al. [16, Proposition 3.2]. Next, the third inequality follows from [38, Lemma 2.2]. Finally, averaging (7) for $i = 1, \ldots, n$ and taking into account that $f(x) = \frac{1}{n} \sum_{i=1}^{n} f_i(x)$, we get (9). ☐

> **Lemma 2.** *Let $f_i(x)$ be generalized smooth with parameters $L_0, L_1 > 0$, and lower bounded by $f_i^{\text{inf}}$, and let $f(x)$ be lower bounded by $f^{\text{inf}}$. Then, for any $x \in \mathbb{R}^d$*
>
> $$\frac{1}{n} \sum_{i=1}^{n} \|\nabla f_i(x)\| \leq 8L_1(f(x) - f^{\text{inf}}) + \frac{8L_1}{n} \sum_{i=1}^{n} (f^{\text{inf}} - f_i^{\text{inf}}) + L_0/L_1. \tag{10}$$

*Proof.* By the $(L_0, L_1)$-smoothness of $f_i(x)$,

$$4(f_i(x) - f_i^{\text{inf}}) \overset{(8)}{\geq} \frac{\|\nabla f_i(x)\|^2}{L_0 + L_1 \|\nabla f_i(x)\|} \geq \begin{cases} \frac{\|\nabla f_i(x)\|^2}{2L_0} & \text{if } \|\nabla f_i(x)\| \leq \frac{L_0}{L_1} \\ \frac{\|\nabla f_i(x)\|}{2L_1} & \text{otherwise.} \end{cases}$$

This condition implies

$$
\begin{aligned}
\|\nabla f_i(x)\| &\leq \max(8L_1(f_i(x) - f_i^{\text{inf}}), L_0/L_1) \\
&\leq 8L_1(f_i(x) - f_i^{\text{inf}}) + L_0/L_1 \\
&\leq 8L_1(f_i(x) - f^{\text{inf}}) + 8L_1(f^{\text{inf}} - f_i^{\text{inf}}) + L_0/L_1.
\end{aligned}
$$

Finally, by the fact that $f(x) = \frac{1}{n}\sum_{i=1}^{n} f_i(x)$,

$$
\frac{1}{n}\sum_{i=1}^{n}\|\nabla f_i(x)\| \leq 8L_1(f(x) - f^{\text{inf}}) + \frac{8L_1}{n}\sum_{i=1}^{n}(f^{\text{inf}} - f_i^{\text{inf}}) + L_0/L_1.
$$

$\square$

**Lemma 3.** *Let $f(x) = \frac{1}{n}\sum_{i=1}^{n} f_i(x)$, where each $f_i(x)$ is generalized smooth with parameters $L_0, L_1 > 0$. Let $x^{k+1} = x^k - \frac{\gamma_k}{\|v^k\|}v^k$ for $\gamma_k > 0$. Then,*

$$
\begin{aligned}
f(x^{k+1}) &\leq f(x^k) - \gamma_k\|\nabla f(x^k)\| + 2\gamma_k\|\nabla f(x^k) - v^k\| \\
&\quad + \frac{\gamma_k^2}{2}\exp(\gamma_k L_1)\left(L_0 + \frac{L_1}{n}\sum_{i=1}^{n}\|\nabla f_i(x^k)\|\right).
\end{aligned}
$$

*Proof.* Let each $f_i(x)$ be generalized smooth with $L_0, L_1 > 0$, and $f(x) = \frac{1}{n}\sum_{i=1}^{n} f_i(x)$. By (9) of Lemma 1, and by the fact that $x^{k+1} = x^k - \frac{\gamma_k}{\|v^k\|}v^k$ for $\gamma_k > 0$,

$$
\begin{aligned}
f(x^{k+1}) &\leq f(x^k) - \frac{\gamma_k}{\|v^k\|}\langle\nabla f(x^k), v^k\rangle + \frac{\gamma_k^2}{2}\exp(\gamma_k L_1)\left(L_0 + \frac{L_1}{n}\sum_{i=1}^{n}\|\nabla f_i(x^k)\|\right) \\
&= f(x^k) - \frac{\gamma_k}{\|v^k\|}\langle\nabla f(x^k) - v^k, v^k\rangle - \gamma_k\|v^k\| \\
&\quad + \frac{\gamma_k^2}{2}\exp(\gamma_k L_1)\left(L_0 + \frac{L_1}{n}\sum_{i=1}^{n}\|\nabla f_i(x^k)\|\right) \\
&\leq f(x^k) + \gamma_k\|\nabla f(x^k) - v^k\| - \gamma_k\|v^k\| \\
&\quad + \frac{\gamma_k^2}{2}\exp(\gamma_k L_1)\left(L_0 + \frac{L_1}{n}\sum_{i=1}^{n}\|\nabla f_i(x^k)\|\right),
\end{aligned}
$$

where we reach the last inequality by Cauchy-Schwarz inequality. Next, since

$$
-\|v^k\| \overset{\text{triangle ineq.}}{\leq} -\|\nabla f(x^k)\| + \|\nabla f(x^k) - v^k\|,
$$

we get

$$
\begin{aligned}
f(x^{k+1}) &\leq f(x^k) - \gamma_k\|\nabla f(x^k)\| + 2\gamma_k\|\nabla f(x^k) - v^k\| \\
&\quad + \frac{\gamma_k^2}{2}\exp(\gamma_k L_1)\left(L_0 + \frac{L_1}{n}\sum_{i=1}^{n}\|\nabla f_i(x^k)\|\right).
\end{aligned}
$$

$\square$

**Lemma 4.** *Let $\{V^k\}_{k\geq 0}, \{W^k\}_{k\geq 0}$ be non-negative sequences satisfying*

$$
V^{k+1} \leq (1 + b_1\exp(L_1\gamma)\gamma^2)V^k - b_2\gamma W^k + b_3\exp(L_1\gamma)\gamma^2,
$$

*for $\gamma, b_1, b_2, b_3 > 0$. Then,*

$$
\min_{k=0,1,\ldots,K} W^k \leq \frac{V^0\exp(b_1\exp(L_1\gamma)\gamma^2(K+1))}{b_2\gamma(K+1)} + \frac{b_3}{b_2}\exp(L_1\gamma)\gamma.
$$

*Proof.* Define $\beta_k = \frac{\beta_{k-1}}{1+b_1\exp(L_1\gamma)\gamma^2}$ for $k = 0, 1, \dots$ and $\beta_{-1} = 1$. Then, we can show that $\beta_k = \frac{1}{(1+b_1\exp(L_1\gamma)\gamma^2)^{k+1}}$ for $k = 0, 1, \dots$, and that

$$\begin{aligned}
\beta_k V^{k+1} &\leq (1+b_1\exp(L_1\gamma)\gamma^2)\beta_k V^k - b_2\gamma\beta_k W^k + b_3\exp(L_1\gamma)\gamma^2\beta_k \\
&= \beta_{k-1}V^k - b_2\gamma\beta_k W^k + b_3\exp(L_1\gamma)\gamma^2\beta_k.
\end{aligned}$$

Therefore,

$$\begin{aligned}
\min_{k=0,1,\dots,K} W^k &\leq \frac{1}{\sum_{k=0}^K \beta_k}\sum_{k=0}^K \beta_k W^k \\
&\leq \frac{\sum_{k=0}^K(\beta_{k-1}V^k - \beta_k V^{k+1})}{b_2\gamma\sum_{k=0}^K \beta_k} + \frac{b_3}{b_2}\exp(L_1\gamma)\gamma \\
&= \frac{\beta_{-1}V^0 - \beta_K V^{k+1}}{b_2\gamma\sum_{k=0}^K \beta_k} + \frac{b_3}{b_2}\exp(L_1\gamma)\gamma.
\end{aligned}$$

By the fact that $\beta_{-1} = 1$, $\beta_K > 0$, and $V^{k+1} \geq 0$,

$$\min_{k=0,1,\dots,K} W^k \leq \frac{V^0}{b_2\gamma\sum_{k=0}^K \beta_k} + \frac{b_3}{b_2}\exp(L_1\gamma)\gamma.$$

Next, since

$$\sum_{k=0}^K \beta_k \geq (K+1)\min_{k=0,1,\dots,K}\beta_k = \frac{K+1}{(1+b_1\exp(L_1\gamma)\gamma^2)^{K+1}},$$

we have

$$\begin{aligned}
\min_{k=0,1,\dots,K} W^k &\leq \frac{V^0(1+b_1\exp(L_1\gamma)\gamma^2)^{K+1}}{b_2\gamma(K+1)} + \frac{b_3}{b_2}\exp(L_1\gamma)\gamma \\
&\overset{1+x\leq\exp(x)}{\leq} \frac{V^0\exp(b_1\exp(L_1\gamma)\gamma^2(K+1))}{b_2\gamma(K+1)} + \frac{b_3}{b_2}\exp(L_1\gamma)\gamma.
\end{aligned}$$

$\square$

# B   Convergence Proof for ||EF21|| (Theorem 1)

In this section, we derive the convergence rate results of ||EF21||. We start with the following lemma technical lemma.

**Lemma 5.** *Let Assumptions 3 and 4 hold. Then, the iterates $\{x^k\}$ generated by ||EF21|| (Algorithm 1) satisfy*

$$
\mathrm{E}\left[\left\|\nabla f_i(x^{k+1}) - g_i^{k+1}\right\|\right] \leq \sqrt{1-\alpha}\mathrm{E}\left[\left\|\nabla f_i(x^k) - g_i^k\right\|\right]
$$
$$
+\sqrt{1-\alpha}\exp(L_1\gamma_k)\gamma_k(L_0 + L_1\mathrm{E}\left[\left\|\nabla f_i(x^k)\right\|\right]). \quad (11)
$$

*Proof.* From the definition of the Euclidean norm, and by taking the expectation conditioned on $x^{k+1}, g_i^k$, and by the update of $g_i^k$ from Algorithm 1

$$
\mathrm{E}\left[\left\|\nabla f_i(x^{k+1}) - g_i^{k+1}\right\| \Big| x^{k+1}, g_i^k\right]
$$
$$
= \mathrm{E}\left[\left\|\nabla f_i(x^{k+1}) - g_i^k - \mathcal{C}^k(\nabla f_i(x^{k+1}) - g_i^k)\right\| \Big| x^{k+1}, g_i^k\right]
$$
$$
\leq \sqrt{\mathrm{E}\left[\left\|\nabla f_i(x^{k+1}) - g_i^k - \mathcal{C}(\nabla f_i(x^{k+1}) - g_i^k)\right\|^2 \Big| x^{k+1}, g_i^k\right]},
$$

where we use the concavity of the square root function, and Jensen's inequality for the concave function, i.e., $\mathrm{E}\left[f(x)\right] \leq f(\mathrm{E}\left[x\right])$ if $f(x)$ is concave. By the $\alpha$-contractive property of compressors in (3), by the fact that $\left\|\nabla f_i(x^{k+1}) - g_i^k\right\|$ is a constant conditioned on $x^{k+1}, g_i^k$, and then by the triangle inequality, we have

$$
\mathrm{E}\left[\left\|\nabla f_i(x^{k+1}) - g_i^{k+1}\right\| \Big| x^{k+1}, g_i^k\right] \leq \sqrt{(1-\alpha)\mathrm{E}\left[\left\|\nabla f_i(x^{k+1}) - g_i^k\right\|^2 \Big| x^{k+1}, g_i^k\right]}
$$
$$
= \sqrt{1-\alpha}\left\|\nabla f_i(x^{k+1}) - g_i^k\right\|
$$
$$
\leq \sqrt{1-\alpha}\left\|\nabla f_i(x^k) - g_i^k\right\| + \sqrt{1-\alpha}\left\|\nabla f_i(x^{k+1}) - \nabla f_i(x^k)\right\|.
$$

By the generalized smoothness of $f_i(x)$ in (2), and by the fact that $x^{k+1} = x^k - \gamma_k\frac{g^k}{\|g^k\|}$,

$$
\mathrm{E}\left[\left\|\nabla f_i(x^{k+1}) - g_i^{k+1}\right\| \Big| x^{k+1}, g_i^k\right] \leq \sqrt{1-\alpha}\left\|\nabla f_i(x^k) - g_i^k\right\|
$$
$$
+\sqrt{1-\alpha}(L_0 + L_1\left\|\nabla f_i(x^k)\right\|)\exp(L_1\gamma_k)\gamma_k.
$$

Let $\gamma_k > 0$ be constants conditioned on $x^{k+1}, g_i^k$. Then, by the tower property, i.e.,

$$
\mathrm{E}\left[\left\|\nabla f_i(x^{k+1}) - g_i^{k+1}\right\|\right] = \mathrm{E}\left[\mathrm{E}\left[\left\|\nabla f_i(x^{k+1}) - g_i^{k+1}\right\| \Big| x^{k+1}, g_i^k\right]\right],
$$

we have

$$
\mathrm{E}\left[\left\|\nabla f_i(x^{k+1}) - g_i^{k+1}\right\|\right] \leq \sqrt{1-\alpha}\mathrm{E}\left[\left\|\nabla f_i(x^k) - g_i^k\right\|\right]
$$
$$
+\sqrt{1-\alpha}\exp(L_1\gamma_k)\gamma_k(L_0 + L_1\mathrm{E}\left[\left\|\nabla f_i(x^k)\right\|\right]).
$$

This concludes the proof. $\qquad\square$

Next, we present the following descent lemma for ||EF21||.

**Lemma 6.** *Let Assumptions 1-4 hold. Then, the iterates $\{x^k\}$ generated by ||EF21|| (Algorithm 1) satisfy*

$$
\mathrm{E}\left[V^{k+1}\right] \leq \mathrm{E}\left[V^k\right] + c_1\gamma_k^2\frac{1}{n}\sum_{i=1}^n\mathrm{E}\left[\left\|\nabla f_i(x^k)\right\|\right] - \gamma_k\mathrm{E}\left[\left\|\nabla f(x^k)\right\|\right] + c_0\gamma_k^2,
$$

*where $V^k := f(x^k) - f^{\inf} + \frac{2\gamma_k}{1-\sqrt{1-\alpha}}\frac{1}{n}\sum_{i=1}^n\left\|\nabla f_i(x^k) - g_i^k\right\|$, and $c_i = \frac{L_i}{2} + 2\frac{\sqrt{1-\alpha}L_i}{1-\sqrt{1-\alpha}}$ for $i = 0, 1$.*

*Proof.* For brevity, let $A_k = \frac{2\gamma_k}{1-\sqrt{1-\alpha}}$. Then, we have $V^k := f(x^k) - f^{\inf} + A_k \frac{1}{n} \sum_{i=1}^n \left\|\nabla f_i(x^k) - v_i^k\right\|$, and from Lemma 3, we derive

$$
\begin{aligned}
\mathrm{E}\left[V^{k+1}\right] \leq{} & \mathrm{E}\left[f(x^k) - f^{\inf}\right] - \gamma_k \mathrm{E}\left[\left\|\nabla f(x^k)\right\|\right] \\
& + \exp(L_1\gamma_k)\gamma_k^2 \frac{L_1}{2n} \sum_{i=1}^n \mathrm{E}\left[\left\|\nabla f_i(x^k)\right\|\right] + \exp(L_1\gamma_k)\gamma_k^2 \frac{L_0}{2} \\
& + 2\gamma_k \mathrm{E}\left[\left\|\nabla f(x^k) - g^k\right\|\right] + A_{k+1}\frac{1}{n} \sum_{i=1}^n \mathrm{E}\left[\left\|\nabla f_i(x^{k+1}) - g_i^{k+1}\right\|\right].
\end{aligned}
$$

Identities $\nabla f(x^k) = \frac{1}{n}\sum_{i=1}^n \nabla f_i(x^k)$ and $g^k = \frac{1}{n}\sum_{i=1}^n g_i^k$ and the triangle inequality imply

$$
\begin{aligned}
\mathrm{E}\left[V^{k+1}\right] \leq{} & \mathrm{E}\left[f(x^k) - f^{\inf}\right] - \gamma_k \mathrm{E}\left[\left\|\nabla f(x^k)\right\|\right] \\
& + \exp(L_1\gamma_k)\gamma_k^2 \frac{L_1}{2n} \sum_{i=1}^n \mathrm{E}\left[\left\|\nabla f_i(x^k)\right\|\right] + \exp(L_1\gamma_k)\gamma_k^2 \frac{L_0}{2} \\
& + 2\gamma_k \frac{1}{n}\sum_{i=1}^n \mathrm{E}\left[\left\|\nabla f_i(x^k) - g_i^k\right\|\right] + A_{k+1}\frac{1}{n} \sum_{i=1}^n \mathrm{E}\left[\left\|\nabla f_i(x^{k+1}) - g_i^{k+1}\right\|\right].
\end{aligned}
$$

Next, we apply (11):

$$
\begin{aligned}
\mathrm{E}\left[V^{k+1}\right] \leq{} & \mathrm{E}\left[f(x^k) - f^{\inf}\right] - \gamma_k \mathrm{E}\left[\left\|\nabla f(x^k)\right\|\right] + \left(\frac{\gamma_k^2}{2} + A_{k+1}\sqrt{1-\alpha}\gamma_k\right)\exp(L_1\gamma_k)L_0 \\
& + \left(\frac{\gamma_k^2}{2} + A_{k+1}\sqrt{1-\alpha}\gamma_k\right)\exp(L_1\gamma_k)L_1\frac{1}{n}\sum_{i=1}^n \mathrm{E}\left[\left\|\nabla f_i(x^k)\right\|\right] \\
& + \left(2\gamma_k + A_{k+1}\sqrt{1-\alpha}\right)\frac{1}{n}\sum_{i=1}^n \mathrm{E}\left[\left\|\nabla f_i(x^k) - g_i^k\right\|\right].
\end{aligned}
$$

If $A_k = \frac{2\gamma_k}{1-\sqrt{1-\alpha}}$, and $\gamma_k$ satisfies $\gamma_{k+1} \leq \gamma_k$, then

$$
2\gamma_k + A_{k+1}\sqrt{1-\alpha} \leq 2\gamma_k + A_k\sqrt{1-\alpha} = A_k.
$$

Therefore,

$$
\begin{aligned}
\mathrm{E}\left[V^{k+1}\right] \leq{} & \mathrm{E}\left[V^k\right] + c_1 \exp(L_1\gamma_k)\gamma_k^2 \frac{1}{n}\sum_{i=1}^n \mathrm{E}\left[\left\|\nabla f_i(x^k)\right\|\right] \\
& - \gamma_k \mathrm{E}\left[\left\|\nabla f(x^k)\right\|\right] + c_0 \exp(L_1\gamma_k)\gamma_k^2,
\end{aligned}
$$

where $c_i = \frac{L_i}{2} + 2\frac{\sqrt{1-\alpha}L_i}{1-\sqrt{1-\alpha}}$ for $i = 0, 1$. $\qquad\square$

## B.1 Proof of Theorem 1

Now, we are ready to prove Theorem 1. From Lemma 6 and 2, and by the fact that $c_1 L_0 / L_1 = c_0$, we have

$$
\begin{aligned}
\mathrm{E}\left[V^{k+1}\right] \leq{} & \mathrm{E}\left[V^k\right] + 8c_1 L_1 \exp(L_1\gamma_k)\gamma_k^2 \mathrm{E}\left[f(x^k) - f^{\inf}\right] \\
& - \gamma_k \mathrm{E}\left[\left\|\nabla f(x^k)\right\|\right] + B\exp(L_1\gamma_k)\gamma_k^2,
\end{aligned}
$$

where $B = 2c_0 + \frac{8c_1 L_1}{n}\sum_{i=1}^n (f^{\inf} - f_i^{\inf})$. Using the fact that $f(x^k) - f^{\inf} \leq V^k$, we derive

$$
\mathrm{E}\left[V^{k+1}\right] \leq (1 + 8c_1 L_1 \exp(L_1\gamma_k)\gamma_k^2)\mathrm{E}\left[V^k\right] - \gamma_k \mathrm{E}\left[\left\|\nabla f(x^k)\right\|\right] + B\exp(L_1\gamma_k)\gamma_k^2.
$$

Applying Lemma 4 with $V^k = \mathrm{E}\left[V^k\right]$, $W^k = \mathrm{E}\left[\left\|\nabla f(x^k)\right\|\right]$, $b_1 = 8c_1 L_1$, $b_2 = 1$, and $b_3 = B$, we get

$$
\min_{k=0,1,\ldots,K} W^k \leq \frac{V^0 \exp(b_1 \exp(L_1\gamma)\gamma^2(K+1))}{b_2\gamma(K+1)} + \frac{b_3}{b_2}\exp(L_1\gamma)\gamma.
$$

Finally, if $\gamma = \frac{\gamma_0}{\sqrt{K+1}}$ with $\gamma_0 > 0$, then $\exp(L_1\gamma_k) \leq \exp(L_1\gamma_0)$, and thus

$$
\min_{k=0,1,\ldots,K} W^k \leq \frac{V^0 \exp(b_1 \exp(L_1\gamma_0)\gamma_0^2)}{b_2\gamma_0\sqrt{K+1}} + \frac{b_3}{b_2}\frac{\gamma_0 \exp(L_1\gamma_0)}{\sqrt{K+1}}.
$$

## C    Discussion on Theorem 1

In this section, we compare the convergence bound between ||EF21|| and EF21 under traditional smoothness. For nonconvex, traditional smooth problems, ||EF21|| from Theorem 1 with $L_1 = 0$ achieves the same $\mathcal{O}(1/\sqrt{K})$ rate in the expectation of gradient norms as EF21 analyzed by Richtárik et al. [8], but with a larger convergence factor. We prove this by assuming $\nabla f_i(x^0) = g_i^0$ for all $i$. That is, Theorem 1 with $L_0 = L$, $L_1 = 0$, $\gamma_0 = \sqrt{(f(x^0) - f^{\inf})/(2b)}$, and $b = \frac{L}{2} + 2\frac{\sqrt{1-\alpha}L}{1-\sqrt{1-\alpha}}$ implies that ||EF21|| achieves

$$
\min_{k=0,1,\ldots,K} \mathrm{E}\left[\left\|\nabla f(x^k)\right\|\right] \leq \frac{1}{\sqrt{K+1}}\left[\frac{f(x^0) - f^{\inf}}{\gamma_0} + 2b\gamma_0\right]
$$

$$
\leq 2\sqrt{L\frac{(1+3\sqrt{1-\alpha})(1+\sqrt{1-\alpha})}{\alpha}}\sqrt{\frac{f(x^0) - f^{\inf}}{K+1}}
$$

$$
\overset{\alpha \geq 0}{\lesssim} 4\sqrt{2}\sqrt{\frac{L}{\alpha}}\sqrt{\frac{f(x^0) - f^{\inf}}{K+1}}.
$$

On the other hand, EF21 attains from Theorem 1 of [8] with $L_i = \tilde{L} = L$ (i.e., $f_i(x)$ has the same smoothness constant as $f(x)$), and $\hat{x}^K$ being chosen from the iterates $x^0, x^1, \ldots, x^K$ uniformly at random

$$
\min_{k=0,1,\ldots,K} \mathrm{E}\left[\left\|\nabla f(x^k)\right\|\right] \leq \mathrm{E}\left[\left\|\nabla f(\hat{x}^K)\right\|\right]
$$

$$
\leq \sqrt{\mathrm{E}\left[\left\|\nabla f(\hat{x}^K)\right\|^2\right]}
$$

$$
\leq \sqrt{2L(1 + \sqrt{\beta/\theta})\frac{f(x^0) - f^{\inf}}{K+1}}
$$

$$
\overset{\sqrt{\beta/\theta} \leq 2/\alpha - 1}{\leq} 2\sqrt{\frac{L}{\alpha}}\sqrt{\frac{f(x^0) - f^{\inf}}{K+1}}.
$$

In conclusion, the convergence bound of ||EF21|| is slower by a factor of $2\sqrt{2}$ than the original EF21 for nonconvex, $L$-smooth problems.

# D   Convergence of ||EF21|| for a Single-node Case

In this section, we provide the convergence of ||EF21|| for a single-node case. In particular, the algorithm enjoys the $\mathcal{O}(1/K)$ convergence up to the error of $\frac{c_0\gamma}{1-c_1\exp(L_1\gamma)\gamma}$. In contrast to Theorem 1 for multi-node ||EF21||, the next result for single-node ||EF21|| applies for any $\gamma_k = \gamma \in (0, 1/(\beta c_1))$ with $\beta \geq 2$, $c_1 = \frac{L_1}{2} + 2\frac{\sqrt{1-\alpha}L_1}{1-\sqrt{1-\alpha}}$, and $\alpha \in (0, 1]$.

**Theorem 3.** *Let Assumptions 1-4 hold. Then, the iterates $\{x^k\}$ generated by ||EF21|| (Algorithm 1) with $n = 1$, $\gamma_k = \gamma = 1/(\beta c_1)$ and $\beta \geq 2$ satisfy*

$$\min_{k=0,1,\dots,K} \mathrm{E}\left[\left\|\nabla f(x^k)\right\|\right] \leq \frac{\mathrm{E}\left[V^0\right] - \mathrm{E}\left[V^{K+1}\right]}{\gamma(1 - c_1\exp(L_1\gamma)\gamma)(K+1)} + \frac{c_0\gamma}{1 - c_1\exp(L_1\gamma)\gamma},$$

*where $V^k = f(x^k) - f^{\inf} + \frac{2\gamma}{1-\sqrt{1-\alpha}}\left\|\nabla f(x^k) - g^k\right\|$, and $c_i = \frac{L_i}{2} + 2\frac{\sqrt{1-\alpha}L_i}{1-\sqrt{1-\alpha}}$ for $i = 0, 1$.*

*Proof.* In the single-node case, Lemma 5 implies

$$
\begin{aligned}
\mathrm{E}\left[\left\|\nabla f(x^{k+1}) - g^{k+1}\right\|\right] &\leq \sqrt{1-\alpha}\mathrm{E}\left[\left\|\nabla f(x^k) - g^k\right\|\right] \\
&\quad + \sqrt{1-\alpha}\exp(L_1\gamma_k)\gamma_k(L_0 + L_1\mathrm{E}\left[\left\|\nabla f(x^k)\right\|\right]). \quad (12)
\end{aligned}
$$

Next, for brevity, let $A_k = \frac{2\gamma_k}{1-\sqrt{1-\alpha}}$. Then, we have $V^k := f(x^k) - f^{\inf} + A_k\frac{1}{n}\sum_{i=1}^n \left\|\nabla f_i(x^k) - g_i^k\right\|$, and from Lemma 3, we derive

$$
\begin{aligned}
\mathrm{E}\left[V^{k+1}\right] &\leq \mathrm{E}\left[f(x^k) - f^{\inf}\right] - \left(\gamma_k - \frac{\gamma_k^2 L_1}{2}\exp(L_1\gamma_k)\right)\mathrm{E}\left[\left\|\nabla f(x^k)\right\|\right] + \frac{\gamma_k^2 L_0}{2}\exp(L_1\gamma_k) \\
&\quad + 2\gamma_k\mathrm{E}\left[\left\|\nabla f(x^k) - g^k\right\|\right] + A_{k+1}\mathrm{E}\left[\left\|\nabla f(x^{k+1}) - g^{k+1}\right\|\right] \\
&\overset{(12)}{\leq} \mathrm{E}\left[f(x^k) - f^{\inf}\right] + \left(2\gamma_k + A_{k+1}\sqrt{1-\alpha}\right)\mathrm{E}\left[\left\|\nabla f(x^k) - g^k\right\|\right] \\
&\quad - \left(\gamma_k - \frac{\gamma_k^2 L_1}{2}\exp(L_1\gamma_k) - A_{k+1}\sqrt{1-\alpha}L_1\gamma_k\exp(L_1\gamma_k)\right)\mathrm{E}\left[\left\|\nabla f(x^k)\right\|\right] \\
&\quad + \frac{\gamma_k^2 L_0}{2}\exp(L_1\gamma_k) + A_{k+1}\sqrt{1-\alpha}L_0\gamma_k\exp(L_1\gamma_k).
\end{aligned}
$$

If $A_k = \frac{2\gamma_k}{1-\sqrt{1-\alpha}}$ and $\gamma_k$ satisfies $\gamma_{k+1} \leq \gamma_k$, then

$$2\gamma_k + A_{k+1}\sqrt{1-\alpha} \leq 2\gamma_k + A_k\sqrt{1-\alpha} = A_k.$$

Therefore,

$$\mathrm{E}\left[V^{k+1}\right] \leq \mathrm{E}\left[V^k\right] - \left(\gamma_k - c_1\exp(L_1\gamma_k)\gamma_k^2\right)\mathrm{E}\left[\left\|\nabla f(x^k)\right\|\right] + c_0\exp(L_1\gamma_k)\gamma_k^2,$$

where $c_i = \frac{L_i}{2} + 2\frac{\sqrt{1-\alpha}L_i}{1-\sqrt{1-\alpha}}$ for $i = 0, 1$.

Finally, taking $\gamma_k = \gamma = 1/(\beta c_1)$ for $\beta \geq 2$, we get $c_1\exp(L_1\gamma)\gamma = \exp(L_1/(\beta c_1))/\beta \leq \exp(2/\beta)/\beta \leq 0.7 < 1$, and

$$\mathrm{E}\left[V^{k+1}\right] \leq \mathrm{E}\left[V^k\right] - \gamma\left(1 - c_1\exp(L_1\gamma)\gamma\right)\mathrm{E}\left[\left\|\nabla f(x^k)\right\|\right] + c_0\gamma^2.$$

Rearranging the terms, we derive

$$
\begin{aligned}
\min_{k=0,1,\dots,K} \mathrm{E}\left[\left\|\nabla f(x^k)\right\|\right] &\leq \frac{1}{K+1}\sum_{k=0}^K \mathrm{E}\left[\left\|\nabla f(x^k)\right\|\right] \\
&\leq \frac{\mathrm{E}\left[V^0\right] - \mathrm{E}\left[V^{K+1}\right]}{\gamma(1 - c_1\exp(L_1\gamma)\gamma)(K+1)} + \frac{c_0\gamma}{1 - c_1\exp(L_1\gamma)\gamma}.
\end{aligned}
$$

Noticing that $V^k \geq 0$, we complete the proof. □

# E   Convergence of ||EF21-SGDM|| (Theorem 2)

In this section, we derive the convergence rate results of ||EF21-SGDM|| . We first introduce auxiliary lemmas in Section E.1, and later prove the convergence theorem (Theorem 2) in Section E.2.

## E.1   Auxiliary Lemmas

Now, we provide useful lemmas for analyzing ||EF21-SGDM||. First, Lemma 7 shows the descent inequality of the normalized gradient descent update under Assumption 3 (generalized smoothness of $f_i$). Second, Lemmas 8 and 9 provide the upper-bound of the Euclidean distance between $v_i^k$ and $g_i^k$, and of the Euclidean distance between $v_i^k$ and $\nabla f_i(x^k)$, respectively.

**Lemma 7.** *Consider the iterates $\{x^k\}$ generated by Algorithm 2. If Assumption 3 holds, then for any $\gamma_k > 0, \eta_k \in [0, 1]$,*

$$
\begin{aligned}
f(x^{k+1}) \quad \leq \quad & f(x^k) - \gamma_k \left\| \nabla f(x^k) \right\| + 2\gamma_k \left\| \nabla f(x^k) - v^k \right\| + 2\gamma_k \left\| v^k - g^k \right\| \\
& + L_0 \gamma_k^2 \exp(\gamma_k L_1) + 4L_1^2 \gamma_k^2 \exp(\gamma_k L_1) \left( f(x^k) - f^{\mathrm{inf}} \right) \\
& + \frac{4L_1^2 \gamma_k^2 \exp(\gamma_k L_1)}{n} \sum_{i=1}^{n} \left( f^{\mathrm{inf}} - f_i^{\mathrm{inf}} \right).
\end{aligned}
$$

*Proof.* Applying the triangle inequality in Lemma 3, i.e., $\left\| \nabla f(x^k) - g^k \right\| \leq \left\| \nabla f(x^k) - v^k \right\| + \left\| v^k - g^k \right\|$, we get

$$
\begin{aligned}
f(x^{k+1}) \quad \leq \quad & f(x^k) - \gamma_k \left\| \nabla f(x^k) \right\| + 2\gamma_k \left\| \nabla f(x^k) - v^k \right\| + 2\gamma_k \left\| v^k - g^k \right\| \\
& + \frac{\gamma_k^2}{2} \exp\left(\gamma_k L_1\right) \left( L_0 + \frac{L_1}{n} \sum_{i=1}^{n} \left\| \nabla f_i(x^k) \right\| \right) \\
\overset{(10)}{\leq} \quad & f(x^k) - \gamma_k \left\| \nabla f(x^k) \right\| + 2\gamma_k \left\| \nabla f(x^k) - v^k \right\| + 2\gamma_k \left\| v^k - g^k \right\| \\
& + L_0 \gamma_k^2 \exp(\gamma_k L_1) + 4L_1^2 \gamma_k^2 \exp(\gamma_k L_1) \left( f(x^k) - f^{\mathrm{inf}} \right) \\
& + \frac{4L_1^2 \gamma_k^2 \exp(\gamma_k L_1)}{n} \sum_{i=1}^{n} \left( f^{\mathrm{inf}} - f_i^{\mathrm{inf}} \right),
\end{aligned}
$$

which concludes the proof. □

**Lemma 8.** *Consider the iterates $\{x^k\}$ generated by Algorithm 2. If Assumptions 3, 4, and 5 hold, then for $\gamma_k > 0, \eta_k \in [0, 1]$, and $k \geq 0$,*

$$
\begin{aligned}
\frac{1}{n} \sum_{i=1}^{n} \mathrm{E}\left[ \left\| v_i^{k+1} - g_i^{k+1} \right\| \right] \leq \quad & \frac{\sqrt{1-\alpha}}{n} \sum_{i=1}^{n} \mathrm{E}\left[ \left\| v_i^k - g_i^k \right\| \right] + \frac{\sqrt{1-\alpha}\eta_{k+1}}{n} \sum_{i=1}^{n} \mathrm{E}\left[ \left\| v_i^k - \nabla f_i(x^k) \right\| \right] \\
& + 8L_1^2 \sqrt{1-\alpha}\eta_{k+1}\gamma_k \exp\left(\gamma_k L_1\right) \mathrm{E}\left[ f(x^k) - f^{\mathrm{inf}} \right] \\
& + \frac{8L_1^2 \sqrt{1-\alpha}\eta_{k+1}\gamma_k \exp\left(\gamma_k L_1\right)}{n} \sum_{i=1}^{n} (f^{\mathrm{inf}} - f_i^{\mathrm{inf}}) \\
& + 2L_0 \sqrt{1-\alpha}\eta_{k+1}\gamma_k \exp\left(\gamma_k L_1\right) + \sqrt{1-\alpha}\eta_{k+1}\sigma.
\end{aligned}
$$

*Proof.* Taking conditional expectation with fixed $\mathcal{F}_{k+1} = \{v_i^{k+1}, x^{k+1}, g_i^k\}$, using the concavity of the squared root of the function, and applying the definition of $g_i^k$ in Algorithm 2, we have

$$
\begin{aligned}
\mathrm{E}\left[\left\|v_i^{k+1} - g_i^{k+1}\right\| \Big| \mathcal{F}_{k+1}\right] &\leq \sqrt{\mathrm{E}\left[\left\|v_i^{k+1} - g_i^{k+1}\right\|^2 \Big| \mathcal{F}_{k+1}\right]} \\
&= \sqrt{\mathrm{E}\left[\left\|v_i^{k+1} - g_i^k - \mathcal{C}^k\left(v_i^{k+1} - g_i^k\right)\right\|^2 \Big| \mathcal{F}_{k+1}\right]} \\
&\overset{(3)}{\leq} \sqrt{\mathrm{E}\left[(1-\alpha)\left\|v_i^{k+1} - g_i^k\right\|^2 \Big| \mathcal{F}_{k+1}\right]}.
\end{aligned}
$$

Next, let $\gamma_k = \gamma > 0$, and $\eta_k = \eta \in [0, 1]$. By the fact that $v_i^{k+1}, g_i^k$ are constants being conditioned on $\mathcal{F}_{k+1}$, and by the triangle inequality,

$$
\begin{aligned}
\mathrm{E}\left[\left\|v_i^{k+1} - g_i^{k+1}\right\| \Big| \mathcal{F}_{k+1}\right] &\leq \sqrt{1-\alpha}\left\|v_i^k - g_i^k\right\| + \sqrt{1-\alpha}\left\|v_i^{k+1} - v_i^k\right\| \\
&= \sqrt{1-\alpha}\left\|v_i^k - g_i^k\right\| + \sqrt{1-\alpha}\eta_{k+1}\left\|\nabla f(x^{k+1}; \xi_i^{k+1}) - v_i^k\right\|.
\end{aligned}
$$

Here, the equality comes from the definition of $v_i^{k+1}$ in Algorithm 2. Next, by the triangle inequality,

$$
\begin{aligned}
\mathrm{E}\left[\left\|v_i^{k+1} - g_i^{k+1}\right\| \Big| \mathcal{F}_{k+1}\right] &\leq \sqrt{1-\alpha}\left\|v_i^k - g_i^k\right\| + \sqrt{1-\alpha}\eta_{k+1}\left\|v_i^k - \nabla f_i(x^k)\right\| \\
&\quad + \sqrt{1-\alpha}\eta_{k+1}\left\|\nabla f_i(x^k) - \nabla f_i(x^{k+1})\right\| \\
&\quad + \sqrt{1-\alpha}\eta_{k+1}\left\|\nabla f_i(x^{k+1}; \xi_i^{k+1}) - \nabla f_i(x^{k+1})\right\| \\
&\overset{(6)}{\leq} \sqrt{1-\alpha}\left\|v_i^k - g_i^k\right\| + \sqrt{1-\alpha}\eta_{k+1}\left\|v_i^k - \nabla f_i(x^k)\right\| \\
&\quad + \sqrt{1-\alpha}\eta_{k+1}\left(L_0 + L_1\left\|\nabla f_i(x^k)\right\|\right)\exp\left(L_1\left\|x^{k+1} - x^k\right\|\right)\left\|x^{k+1} - x^k\right\| \\
&\quad + \sqrt{1-\alpha}\eta_{k+1}\left\|\nabla f(x^{k+1}; \xi_i^{k+1}) - \nabla f(x^{k+1})\right\|.
\end{aligned}
$$

Next, using $x^{k+1} - x^k = -\gamma_k \frac{g^k}{\|g^k\|}$, and taking the expectation, we obtain

$$
\begin{aligned}
\mathrm{E}\left[\left\|v_i^{k+1} - g_i^{k+1}\right\|\right] &\leq \sqrt{1-\alpha}\mathrm{E}\left[\left\|v_i^k - g_i^k\right\|\right] + \sqrt{1-\alpha}\eta_{k+1}\mathrm{E}\left[\left\|v_i^k - \nabla f_i(x^k)\right\|\right] \\
&\quad + \sqrt{1-\alpha}\eta_{k+1}\gamma_k\exp\left(\gamma_k L_1\right)\left(L_0 + L_1\mathrm{E}\left[\left\|\nabla f_i(x^k)\right\|\right]\right) \\
&\quad + \sqrt{1-\alpha}\eta_{k+1}\mathrm{E}\left[\left\|\nabla f_i(x^{k+1}; \xi_i^{k+1}) - \nabla f_i(x^{k+1})\right\|\right].
\end{aligned}
$$

Finally, since

$$
\begin{aligned}
\mathrm{E}\left[\left\|\nabla f_i(x^{k+1}; \xi_i^{k+1}) - \nabla f_i(x^{k+1})\right\|\right] &\leq \sqrt{\mathrm{E}\left[\left\|\nabla f_i(x^{k+1}; \xi_i^{k+1}) - \nabla f_i(x^{k+1})\right\|^2\right]} \\
&\overset{(4)}{\leq} \sigma,
\end{aligned}
$$

we derive

$$\frac{1}{n}\sum_{i=1}^{n}\mathrm{E}\left[\|v_i^{k+1}-g_i^{k+1}\|\right] \quad \leq \quad \frac{\sqrt{1-\alpha}}{n}\sum_{i=1}^{n}\mathrm{E}\left[\|v_i^{k}-g_i^{k}\|\right]$$

$$+\frac{\sqrt{1-\alpha}\eta_{k+1}}{n}\sum_{i=1}^{n}\mathrm{E}\left[\|v_i^{k}-\nabla f_i(x^{k})\|\right]$$

$$+\sqrt{1-\alpha}\eta_{k+1}\gamma_k\exp\left(\gamma_k L_1\right)\left(L_0+L_1\frac{1}{n}\sum_{i=1}^{n}\mathrm{E}\left[\|\nabla f_i(x^{k})\|\right]\right)$$

$$+\sqrt{1-\alpha}\eta_{k+1}\sigma$$

$$\overset{(10)}{\leq} \quad \frac{\sqrt{1-\alpha}}{n}\sum_{i=1}^{n}\mathrm{E}\left[\|v_i^{k}-g_i^{k}\|\right]$$

$$+\frac{\sqrt{1-\alpha}\eta_{k+1}}{n}\sum_{i=1}^{n}\mathrm{E}\left[\|v_i^{k}-\nabla f_i(x^{k})\|\right]$$

$$+8L_1^2\sqrt{1-\alpha}\eta_{k+1}\gamma_k\exp\left(\gamma_k L_1\right)\mathrm{E}\left[f(x^{k})-f^{\mathrm{inf}}\right]$$

$$+\frac{8L_1^2\sqrt{1-\alpha}\eta_{k+1}\gamma_k\exp\left(\gamma_k L_1\right)}{n}\sum_{i=1}^{n}(f^{\mathrm{inf}}-f_i^{\mathrm{inf}})$$

$$+2L_0\sqrt{1-\alpha}\eta_{k+1}\gamma_k\exp\left(\gamma_k L_1\right)+\sqrt{1-\alpha}\eta_{k+1}\sigma.$$

This concludes the proof. $\qquad\square$

**Lemma 9.** *Consider the iterates $\{x^k\}$ generated by Algorithm 2. If Assumptions 3, and 5 hold, then for any $\gamma_k \equiv \gamma > 0$, $\eta_k \equiv \eta$, and $k \geq 0$,*

$$\mathrm{E}\left[\|v^{k}-\nabla f(x^{k})\|\right] \quad \leq \quad (1-\eta)^{k}\mathrm{E}\left[\|v^{0}-\nabla f(x^{0})\|\right]+\frac{\sqrt{\eta}\sigma}{\sqrt{n}}+\frac{2L_0\gamma\exp\left(\gamma L_1\right)}{\eta}$$

$$+8L_1^2\gamma\exp\left(\gamma L_1\right)\sum_{t=0}^{k-1}(1-\eta)^{k-t}\mathrm{E}\left[f(x^{t})-f^{\mathrm{inf}}\right]$$

$$+\frac{8L_1^2\gamma\exp\left(\gamma L_1\right)}{\eta n}\sum_{i=1}^{n}\left(f^{\mathrm{inf}}-f_i^{\mathrm{inf}}\right). \tag{13}$$

*In addition, for any $k \geq 0$,*

$$\frac{1}{n}\sum_{i=1}^{n}\mathrm{E}\left[\|v_i^{k+1}-\nabla f_i(x^{k+1})\|\right] \quad \leq \quad \frac{1-\eta}{n}\sum_{i=1}^{n}\mathrm{E}\left[\|v_i^{k}-\nabla f_i(x^{k})\|\right]+\eta\sigma+2L_0\gamma\exp\left(\gamma L_1\right)$$

$$+8L_1^2\gamma\exp(\gamma L_1)\mathrm{E}\left[f(x^{k})-f^{\mathrm{inf}}\right]$$

$$+\frac{8L_1^2\gamma\exp(\gamma L_1)}{n}\sum_{i=1}^{n}\left(f^{\mathrm{inf}}-f_i^{\mathrm{inf}}\right). \tag{14}$$

*Proof.* We prove the result using the arguments similar to those given in the proof of Theorem 1 from Cutkosky and Mehta [50]. From the definition of $v_i^{k+1}$, we have the following recursion for any $k \geq 0$:

$$\begin{aligned}
v_i^{k+1} &= (1-\eta)v_i^{k}+\eta\nabla f_i(x^{k+1};\xi_i^{k+1}) \\
&= \nabla f_i(x^{k+1})+(1-\eta)(v_i^{k}-\nabla f_i(x^{k}))+(1-\eta)(\nabla f_i(x^{k})-\nabla f_i(x^{k+1})) \\
&\quad +\eta(\nabla f_i(x^{k+1};\xi_i^{k+1})-\nabla f_i(x^{k+1})).
\end{aligned}$$

Next, from the recursion of $v_i^{k+1}$, we obtain the following recursion for $k \geq 0$:

$$H_i^{k+1} \quad = \quad (1-\eta)H_i^{k}+(1-\eta)G_i^{k}+\eta U_i^{k+1}, \tag{15}$$

where
$$U_i^{k+1} = \nabla f_i(x^{k+1}; \xi_i^{k+1}) - \nabla f_i(x^{k+1}), \quad G_i^k = \nabla f_i(x^k) - \nabla f_i(x^{k+1}), \quad H_i^k = v_i^k - \nabla f_i(x^k),$$

$$U^{k+1} = \frac{1}{n}\sum_{i=1}^n U_i^{k+1}, \quad G^k = \frac{1}{n}\sum_{i=1}^n G_i^k, \quad \text{and} \quad H^k = \frac{1}{n}\sum_{i=1}^n H_i^k.$$

Unrolling the recursion for $H_i^k$, we derive

$$H_i^{k+1} = (1-\eta)^{k+1}H_i^0 + \sum_{t=0}^k (1-\eta)^{k-t+1}G_i^t + \eta\sum_{t=0}^k (1-\eta)^{k-t}U_i^{t+1}.$$

Averaging the above inequality, we get

$$H^{k+1} = (1-\eta)^{k+1}H^0 + \sum_{t=0}^k (1-\eta)^{k-t+1}G^t + \eta\sum_{t=0}^k (1-\eta)^{k-t}U^{t+1}.$$

Next, taking the Euclidean norm, using the triangle inequality, and then taking the expectation, we obtain

$$\mathrm{E}\left[\left\|H^{k+1}\right\|\right] \leq (1-\eta)^{k+1}\mathrm{E}\left[\left\|H^0\right\|\right] + \underbrace{\sum_{t=0}^k (1-\eta)^{k-t+1}\mathrm{E}\left[\left\|G^t\right\|\right]}_{=:\mathcal{A}_1}$$

$$+ \eta\,\mathrm{E}\underbrace{\left[\left\|\sum_{t=0}^k (1-\eta)^{k-t}U^{t+1}\right\|\right]}_{=:\mathcal{A}_2}. \tag{16}$$

To bound $\mathrm{E}\left[\left\|H^{k+1}\right\|\right]$, we need to bound the expectation of the last two terms. First, we bound term $\mathcal{A}_1$. Using the fact that $\|G^t\| \leq \frac{1}{n}\sum_{i=1}^n \|G_i^t\|$, and the definition of $G_i^t$, we obtain

$$\mathcal{A}_1 \leq \frac{1}{n}\sum_{i=1}^n\sum_{t=0}^k (1-\eta)^{k-t+1}\mathrm{E}\left[\left\|\nabla f_i(x^t) - \nabla f_i(x^{t+1})\right\|\right]$$

$$\overset{(6)}{\leq} \frac{1}{n}\sum_{i=1}^n\sum_{t=0}^k (1-\eta)^{k-t+1}\mathrm{E}\left[L_0\exp\left(L_1\left\|x^{t+1}-x^t\right\|\right)\left\|x^{t+1}-x^t\right\|\right]$$

$$+ \frac{1}{n}\sum_{i=1}^n\sum_{t=0}^k (1-\eta)^{k-t+1}\mathrm{E}\left[L_1\left\|\nabla f_i(x^t)\right\|\exp\left(L_1\left\|x^{t+1}-x^t\right\|\right)\left\|x^{t+1}-x^t\right\|\right]$$

$$= \sum_{t=0}^k (1-\eta)^{k-t+1}\gamma\exp(\gamma L_1)L_0 + \frac{L_1}{n}\sum_{i=1}^n\sum_{t=0}^k (1-\eta)^{k-t+1}\gamma\exp(\gamma L_1)\mathrm{E}\left[\left\|\nabla f_i(x^t)\right\|\right]$$

$$\overset{(10)}{\leq} 2L_0\gamma\exp(\gamma L_1)\sum_{t=0}^k (1-\eta)^{k-t+1} + 8L_1^2\gamma\exp(\gamma L_1)\sum_{t=0}^k (1-\eta)^{k-t+1}\mathrm{E}\left[f(x^t)-f^{\inf}\right]$$

$$+ \frac{8L_1^2\gamma\exp(\gamma L_1)}{n}\sum_{i=1}^n\sum_{t=0}^k (1-\eta)^{k-t+1}\left(f^{\inf}-f_i^{\inf}\right)$$

$$\leq 2L_0\gamma\exp(\gamma L_1)\sum_{t=0}^\infty (1-\eta)^t + 8L_1^2\gamma\exp(\gamma L_1)\sum_{t=0}^k (1-\eta)^{k-t+1}\mathrm{E}\left[f(x^t)-f^{\inf}\right]$$

$$+ \frac{8L_1^2\gamma\exp(\gamma L_1)}{n}\sum_{i=1}^n \left(f^{\inf}-f_i^{\inf}\right)\sum_{t=0}^\infty (1-\eta)^t$$

$$= \frac{2L_0\gamma\exp(\gamma L_1)}{\eta} + 8L_1^2\gamma\exp(\gamma L_1)\sum_{t=0}^k (1-\eta)^{k-t+1}\mathrm{E}\left[f(x^t)-f^{\inf}\right]$$

$$+ \frac{8L_1^2\gamma\exp(\gamma L_1)}{\eta n}\sum_{i=1}^n \left(f^{\inf}-f_i^{\inf}\right).$$

Next, we bound term $\mathcal{A}_2$. Jensen's inequality and the tower property of the conditional expectation imply

$$\mathcal{A}_2 \leq \sqrt{\mathrm{E}\left[\left\|\sum_{t=0}^{k}(1-\eta)^{k-t}U^{t+1}\right\|^2\right]} = \sqrt{\sum_{t=0}^{k}(1-\eta)^{2(k-t)}\mathrm{E}\left[\left\|U^{t+1}\right\|^2\right]}.$$

Moreover, due to independence of $\{\xi_i^t\}_{i=1}^n$, we have

$$\mathcal{A}_2 \leq \sqrt{\sum_{t=0}^{k}\frac{(1-\eta)^{2(k-t)}}{n^2}\sum_{i=1}^{n}\mathrm{E}\left[\left\|U_i^{t+1}\right\|^2\right]} \overset{(4)}{\leq} \sqrt{\sum_{t=0}^{k}(1-\eta)^{2(k-t)}\frac{\sigma^2}{n}}$$

$$\leq \frac{\sigma}{\sqrt{n}}\sqrt{\sum_{t=0}^{\infty}(1-\eta)^{2t}} = \frac{\sigma}{\sqrt{n\eta(2-\eta)}} \overset{\eta\in[0,1]}{\leq} \frac{\sigma}{\sqrt{n\eta}}.$$

Therefore, plugging the derived upper-bounds for $\mathcal{A}_1$, and for $\mathcal{A}_2$ into (16), we obtain

$$\mathrm{E}\left[\left\|H^{k+1}\right\|\right] \leq (1-\eta)^{k+1}\mathrm{E}\left[\left\|H^0\right\|\right] + \frac{2L_0\gamma\exp\left(\gamma L_1\right)}{\eta}$$

$$+8L_1^2\gamma\exp\left(\gamma L_1\right)\sum_{t=0}^{k}(1-\eta)^{k-t+1}\mathrm{E}\left[f(x^t) - f^{\inf}\right]$$

$$+\frac{8L_1^2\gamma\exp\left(\gamma L_1\right)}{\eta n}\sum_{i=1}^{n}\left(f^{\inf} - f_i^{\inf}\right) + \frac{\sqrt{\eta}\sigma}{\sqrt{n}},$$

which is equivalent to (13).

To derive (14), we make a step back to the recursion from (15), which implies

$$\frac{1}{n}\sum_{i=1}^{n}\mathrm{E}\left[\left\|H_i^{k+1}\right\|\right] \leq \frac{1-\eta}{n}\sum_{i=1}^{n}\mathrm{E}\left[\left\|H_i^k\right\|\right] + \underbrace{\frac{1-\eta}{n}\sum_{i=1}^{n}\mathrm{E}\left[\left\|G_i^k\right\|\right]}_{=:\mathcal{B}_1}$$

$$+\underbrace{\frac{\eta}{n}\sum_{i=1}^{n}\mathrm{E}\left[\left\|U_i^{k+1}\right\|\right]}_{=:\mathcal{B}_2}. \tag{17}$$

Next, we derive the upper bounds for $\mathcal{B}_1$ and $\mathcal{B}_2$. For $\mathcal{B}_1$, we have

$$\mathcal{B}_1 = \frac{1-\eta}{n}\sum_{i=1}^{n}\mathrm{E}\left[\left\|\nabla f_i(x^k) - \nabla f_i(x^{k+1})\right\|\right]$$

$$\overset{(6)}{\leq} \frac{1-\eta}{n}\sum_{i=1}^{n}\mathrm{E}\left[\left(L_0 + L_1\left\|\nabla f_i(x^k)\right\|\right)\exp\left(L_1\left\|x^k - x^{k+1}\right\|\right)\left\|x^k - x^{k+1}\right\|\right]$$

$$= (1-\eta)L_0\gamma\exp(\gamma L_1) + \frac{(1-\eta)L_1\gamma\exp(\gamma L_1)}{n}\sum_{i=1}^{n}\mathrm{E}\left[\left\|\nabla f_i(x^k)\right\|\right]$$

$$\overset{(10)}{\leq} 2(1-\eta)L_0\gamma\exp(\gamma L_1) + 8(1-\eta)L_1^2\gamma\exp(\gamma L_1)\mathrm{E}\left[f(x^k) - f^{\inf}\right]$$

$$+\frac{8(1-\eta)L_1^2\gamma\exp(\gamma L_1)}{n}\sum_{i=1}^{n}\left(f^{\inf} - f_i^{\inf}\right),$$

and for $\mathcal{B}_2$, we obtain

$$\mathcal{B}_2 = \frac{\eta}{n}\sum_{i=1}^{n}\mathrm{E}\left[\left\|\nabla f_i(x^{k+1};\xi_i^{k+1}) - \nabla f_i(x^{k+1})\right\|\right] \overset{(4)}{\leq} \eta\sigma.$$

Plugging the derived upper bounds for $\mathcal{B}_1$ and $\mathcal{B}_2$ into (17) and using $1 - \eta \leq 1$, we get

$$
\begin{aligned}
\frac{1}{n} \sum_{i=1}^{n} \mathrm{E}\left[\left\|H_i^{k+1}\right\|\right] \quad \leq \quad & \frac{1-\eta}{n} \sum_{i=1}^{n} \mathrm{E}\left[\left\|H_i^k\right\|\right] + 2L_0 \gamma \exp(\gamma L_1) \\
& + 8L_1^2 \gamma \exp(\gamma L_1) \mathrm{E}\left[f(x^k) - f^{\mathrm{inf}}\right] \\
& + \frac{8L_1^2 \gamma \exp(\gamma L_1)}{n} \sum_{i=1}^{n}\left(f^{\mathrm{inf}} - f_i^{\mathrm{inf}}\right) + \eta \sigma,
\end{aligned}
$$

which is equivalent to (14).

$\square$

### E.2 Proof of Theorem 2

Now, we are ready to prove Theorem 2. For convenience, we introduce new notation:

$$
\delta^k := \mathrm{E}\left[f(x^k) - f^{\mathrm{inf}}\right], \quad A_k := \frac{1}{n} \sum_{i=1}^{n} \mathrm{E}\left[\left\|v_i^k - g_i^k\right\|\right], \quad B_k := \mathrm{E}\left[\left\|v^k - \nabla f(x^k)\right\|\right],
$$

$$
C_k := \frac{1}{n} \sum_{i=1}^{n} \mathrm{E}\left[\left\|v_i^k - \nabla f_i(x^k)\right\|\right], \quad \delta^{\mathrm{inf}} := \frac{1}{n} \sum_{i=1}^{n}(f^{\mathrm{inf}} - f_i^{\mathrm{inf}}).
$$

Using the new notation and noticing that $\mathrm{E}\left[\left\|v^k - g^k\right\|\right] \leq A_k$, we rewrite the results of Lemmas 7, 8, and 9 as

$$
\begin{aligned}
\delta^{k+1} \quad \leq \quad & \left(1 + 4L_1^2 \gamma^2 \exp(L_1 \gamma)\right) \delta^k + 2\gamma A_k + 2\gamma B_k - \gamma \mathrm{E}\left[\left\|\nabla f(x^k)\right\|\right] \\
& + \gamma^2 \exp(L_1 \gamma)\left(L_0 + 4L_1^2 \delta^{\mathrm{inf}}\right), \\
A_{k+1} \quad \leq \quad & \sqrt{1-\alpha} A_k + \eta\sqrt{1-\alpha} C_k + 8L_1^2 \sqrt{1-\alpha}\eta\gamma \exp\left(\gamma L_1\right) \delta^k \\
& + 2\sqrt{1-\alpha}\eta\gamma \exp\left(\gamma L_1\right)\left(L_0 + 4L_1^2 \delta^{\mathrm{inf}}\right) + \sqrt{1-\alpha}\eta\sigma, \\
B_k \quad \leq \quad & (1-\eta)^k B_0 + \frac{\sqrt{\eta}\sigma}{\sqrt{n}} + \frac{2\gamma \exp\left(L_1 \gamma\right)}{\eta}\left(L_0 + 4L_1^2 \delta^{\mathrm{inf}}\right) \\
& + 8L_1^2 \gamma \exp\left(L_1 \gamma\right) \sum_{t=0}^{k-1}(1-\eta)^{k-t}\delta^t, \\
C_{k+1} \quad \leq \quad & (1-\eta)C_k + 8L_1^2 \gamma \exp\left(L_1 \gamma\right) \delta^k + \eta\sigma + 2\gamma \exp\left(\gamma L_1\right)\left(L_0 + 4L_1^2 \delta^{\mathrm{inf}}\right).
\end{aligned}
$$

Moreover, since $\gamma = \frac{\gamma_0}{(K+1)^{3/4}}$ with $\gamma_0 \leq \frac{1}{2L_1}$, we have $\exp(L_1 \gamma) \leq \exp(L_1 \gamma_0) \leq 2$ and the above inequalities can be further simplified as

$$
\begin{aligned}
\delta^{k+1} \quad \leq \quad & \left(1 + 8L_1^2 \gamma^2\right) \delta^k + 2\gamma A_k + 2\gamma B_k - \gamma \mathrm{E}\left[\left\|\nabla f(x^k)\right\|\right] + 2\gamma^2\left(L_0 + 4L_1^2 \delta^{\mathrm{inf}}\right), \quad (18) \\
A_{k+1} \quad \leq \quad & \sqrt{1-\alpha} A_k + \eta\sqrt{1-\alpha} C_k + 16L_1^2 \sqrt{1-\alpha}\eta\gamma\delta^k \\
& + 4\sqrt{1-\alpha}\eta\gamma\left(L_0 + 4L_1^2 \delta^{\mathrm{inf}}\right) + \sqrt{1-\alpha}\eta\sigma, \quad (19) \\
B_k \quad \leq \quad & (1-\eta)^k B_0 + \frac{\sqrt{\eta}\sigma}{\sqrt{n}} + \frac{4\gamma}{\eta}\left(L_0 + 4L_1^2 \delta^{\mathrm{inf}}\right) + 16L_1^2 \gamma \sum_{t=0}^{k-1}(1-\eta)^{k-t}\delta^t, \quad (20) \\
C_{k+1} \quad \leq \quad & (1-\eta)C_k + 16L_1^2 \gamma\delta^k + \eta\sigma + 4\gamma\left(L_0 + 4L_1^2 \delta^{\mathrm{inf}}\right). \quad (21)
\end{aligned}
$$

Next, we introduce the Lyapunov function $V_k$ defined for any $k \geq 0$ as

$$
V_k = \delta^k + aA_k + cC_k,
$$

where $a := \frac{2\gamma}{1-\sqrt{1-\alpha}}$ and $c := a\sqrt{1-\alpha}$. Then, using (18), (19), (21), we get

$$
\begin{aligned}
V_{k+1} \quad \leq \quad & \left(1 + 8L_1^2 \gamma^2\right) \delta^k + 2\gamma A_k + 2\gamma B_k - \gamma \mathrm{E}\left[\left\|\nabla f(x^k)\right\|\right] + 2\gamma^2\left(L_0 + 4L_1^2 \delta^{\mathrm{inf}}\right) \\
& + a\left(\sqrt{1-\alpha} A_k + \eta\sqrt{1-\alpha} C_k + 16L_1^2 \sqrt{1-\alpha}\eta\gamma\delta^k\right) \\
& + a\left(4\sqrt{1-\alpha}\eta\gamma\left(L_0 + 4L_1^2 \delta^{\mathrm{inf}}\right) + \sqrt{1-\alpha}\eta\sigma\right) \\
& + c\left((1-\eta)C_k + 16L_1^2 \gamma\delta^k + \eta\sigma + 4\gamma\left(L_0 + 4L_1^2 \delta^{\mathrm{inf}}\right)\right).
\end{aligned}
$$

To proceed, we rearrange the terms:

$$
\begin{aligned}
V_{k+1} \quad &\leq \quad \left(1 + 8L_1^2\gamma^2 + 16aL_1^2\sqrt{1-\alpha}\,\eta\gamma + 16cL_1^2\gamma\right)\delta^k + \left(\frac{2\gamma}{a} + \sqrt{1-\alpha}\right)aA_k \\
&\quad + \left(\frac{a\eta\sqrt{1-\alpha}}{c} + 1 - \eta\right)cC_k + 2\gamma B_k - \gamma\mathrm{E}\left[\left\|\nabla f(x^k)\right\|\right] \\
&\quad + \left(2\gamma^2 + 4a\sqrt{1-\alpha}\,\eta\gamma + 4c\gamma\right)\left(L_0 + 4L_1^2\delta^{\mathrm{inf}}\right) + \eta\left(a\sqrt{1-\alpha} + c\right)\sigma \\
&\overset{\substack{c=a\sqrt{1-\alpha},\\ \eta\leq 1}}{\leq} \quad \left(1 + 8L_1^2\gamma^2 + 32aL_1^2\sqrt{1-\alpha}\,\gamma\right)\delta^k + \left(\frac{2\gamma}{a} + \sqrt{1-\alpha}\right)aA_k + cC_k \\
&\quad + 2\gamma B_k - \gamma\mathrm{E}\left[\left\|\nabla f(x^k)\right\|\right] \\
&\quad + \left(2\gamma^2 + 8a\sqrt{1-\alpha}\,\gamma\right)\left(L_0 + 4L_1^2\delta^{\mathrm{inf}}\right) + 2\eta a\sqrt{1-\alpha}\,\sigma.
\end{aligned}
$$

Since $a = \frac{2\gamma}{1-\sqrt{1-\alpha}}$, we have $\frac{2\gamma}{a} + \sqrt{1-\alpha} = 1$ and

$$
\begin{aligned}
V_{k+1} \quad &\leq \quad \left(1 + 8L_1^2\gamma^2 + \frac{64L_1^2\gamma^2\sqrt{1-\alpha}}{1 - \sqrt{1-\alpha}}\right)\delta^k + aA_k + cC_k + 2\gamma B_k - \gamma\mathrm{E}\left[\left\|\nabla f(x^k)\right\|\right] \\
&\quad + \left(2\gamma^2 + \frac{16\gamma^2\sqrt{1-\alpha}}{1 - \sqrt{1-\alpha}}\right)\left(L_0 + 4L_1^2\delta^{\mathrm{inf}}\right) + \frac{4\gamma\eta\sqrt{1-\alpha}\,\sigma}{1 - \sqrt{1-\alpha}} \\
&\leq \quad \left(1 + \frac{64L_1^2\gamma^2}{1 - \sqrt{1-\alpha}}\right)V_k + 2\gamma B_k - \gamma\mathrm{E}\left[\left\|\nabla f(x^k)\right\|\right] \\
&\quad + \frac{16\gamma^2\left(L_0 + 4L_1^2\delta^{\mathrm{inf}}\right)}{1 - \sqrt{1-\alpha}} + \frac{4\gamma\eta\sqrt{1-\alpha}\,\sigma}{1 - \sqrt{1-\alpha}}.
\end{aligned}
$$

Next, we bound $B_k$ using (20) and $\delta^k \leq V_k$:

$$
\begin{aligned}
V_{k+1} \quad &\leq \quad \left(1 + \frac{64L_1^2\gamma^2}{1 - \sqrt{1-\alpha}}\right)V_k + 32L_1^2\gamma^2\sum_{t=0}^{k-1}(1-\eta)^{k-t}V_t + 2\gamma(1-\eta)^k B_0 - \gamma\mathrm{E}\left[\left\|\nabla f(x^k)\right\|\right] \\
&\quad + \left(\frac{16\gamma^2}{1 - \sqrt{1-\alpha}} + \frac{8\gamma^2}{\eta}\right)\left(L_0 + 4L_1^2\delta^{\mathrm{inf}}\right) + \left(\frac{4\gamma\eta\sqrt{1-\alpha}}{1 - \sqrt{1-\alpha}} + \frac{2\gamma\sqrt{\eta}}{\sqrt{n}}\right)\sigma.
\end{aligned}
$$

Summing up the above inequality with weights $\beta_k := \left(1 + \frac{64L_1^2\gamma^2}{1-\sqrt{1-\alpha}} + \frac{32L_1^2\gamma^2}{\eta}\right)^{-(k+1)}$ for $k = 0, \ldots, K$ and denoting $S_K := \sum_{k=0}^{K}\beta_k$ and $\beta_{-1} := 1$, we get

$$
\begin{aligned}
\sum_{k=0}^{K}\beta_k V_{k+1} \quad &\leq \quad \sum_{k=0}^{K}\left(1 + \frac{64L_1^2\gamma^2}{1 - \sqrt{1-\alpha}}\right)\beta_k V_k + 32L_1^2\gamma^2\sum_{k=0}^{K}\beta_k\sum_{t=0}^{k-1}(1-\eta)^{k-t}V_t \\
&\quad + 2\gamma B_0\sum_{k=0}^{K}(1-\eta)^k\beta_k - \gamma\sum_{k=0}^{K}\beta_k\mathrm{E}\left[\left\|\nabla f(x^k)\right\|\right] \\
&\quad + S_K\left(\frac{16\gamma^2}{1 - \sqrt{1-\alpha}} + \frac{8\gamma^2}{\eta}\right)\left(L_0 + 4L_1^2\delta^{\mathrm{inf}}\right) + S_K\left(\frac{4\gamma\eta\sqrt{1-\alpha}}{1 - \sqrt{1-\alpha}} + \frac{2\gamma\sqrt{\eta}}{\sqrt{n}}\right)\sigma.
\end{aligned}
$$

By definition of $\beta_k$, we have $\beta_k \leq \beta_{k-1}$ and, in particular, $\beta_k \leq 1$ for all $k \geq 0$. Using these inequalities, we continue the derivation as follows:

$$
\begin{aligned}
\sum_{k=0}^{K} \beta_k V_{k+1} \quad \leq \quad & \sum_{k=0}^{K} \left(1 + \frac{64 L_1^2 \gamma^2}{1 - \sqrt{1-\alpha}}\right) \beta_k V_k + 32 L_1^2 \gamma^2 \sum_{k=0}^{K} \sum_{t=0}^{k-1} (1-\eta)^{k-t} \beta_t V_t \\
& + 2\gamma B_0 \sum_{k=0}^{K} (1-\eta)^k - \gamma \sum_{k=0}^{K} \beta_k \mathrm{E}\left[\|\nabla f(x^k)\|\right] \\
& + S_K \left(\frac{16\gamma^2}{1 - \sqrt{1-\alpha}} + \frac{8\gamma^2}{\eta}\right) \left(L_0 + 4 L_1^2 \delta^{\mathrm{inf}}\right) + S_K \left(\frac{4\gamma\eta\sqrt{1-\alpha}}{1 - \sqrt{1-\alpha}} + \frac{2\gamma\sqrt{\eta}}{\sqrt{n}}\right) \sigma \\
\leq \quad & \sum_{k=0}^{K} \left(1 + \frac{64 L_1^2 \gamma^2}{1 - \sqrt{1-\alpha}}\right) \beta_k V_k + 32 L_1^2 \gamma^2 \left(\sum_{t=0}^{\infty} (1-\eta)^t\right) \left(\sum_{k=0}^{K} \beta_k V_k\right) \\
& + 2\gamma B_0 \sum_{k=0}^{\infty} (1-\eta)^k - \gamma S_K \min_{k=0,\ldots,K} \mathrm{E}\left[\|\nabla f(x^k)\|\right] \\
& + S_K \left(\frac{16\gamma^2}{1 - \sqrt{1-\alpha}} + \frac{8\gamma^2}{\eta}\right) \left(L_0 + 4 L_1^2 \delta^{\mathrm{inf}}\right) + S_K \left(\frac{4\gamma\eta\sqrt{1-\alpha}}{1 - \sqrt{1-\alpha}} + \frac{2\gamma\sqrt{\eta}}{\sqrt{n}}\right) \sigma \\
= \quad & \sum_{k=0}^{K} \underbrace{\left(1 + \frac{64 L_1^2 \gamma^2}{1 - \sqrt{1-\alpha}} + \frac{32 L_1^2 \gamma^2}{\eta}\right) \beta_k}_{=\beta_{k-1}} V_k + \frac{2\gamma B_0}{\eta} - \gamma S_K \min_{k=0,\ldots,K} \mathrm{E}\left[\|\nabla f(x^k)\|\right] \\
& + S_K \left(\frac{16\gamma^2}{1 - \sqrt{1-\alpha}} + \frac{8\gamma^2}{\eta}\right) \left(L_0 + 4 L_1^2 \delta^{\mathrm{inf}}\right) + S_K \left(\frac{4\gamma\eta\sqrt{1-\alpha}}{1 - \sqrt{1-\alpha}} + \frac{2\gamma\sqrt{\eta}}{\sqrt{n}}\right) \sigma.
\end{aligned}
$$

Rearranging the terms and dividing both sides of the above inequality by $\gamma S_K$, we obtain

$$
\begin{aligned}
\min_{k=0,\ldots,K} \mathrm{E}\left[\|\nabla f(x^k)\|\right] \quad \leq \quad & \frac{1}{\gamma S_K} \sum_{k=0}^{K} (\beta_{k-1} V_k - \beta_k V_{k+1}) + \frac{2 B_0}{\eta S_K} \\
& + \left(\frac{16\gamma}{1 - \sqrt{1-\alpha}} + \frac{8\gamma}{\eta}\right) \left(L_0 + 4 L_1^2 \delta^{\mathrm{inf}}\right) + \left(\frac{4\eta\sqrt{1-\alpha}}{1 - \sqrt{1-\alpha}} + \frac{2\sqrt{\eta}}{\sqrt{n}}\right) \sigma \\
\leq \quad & \frac{V_0}{\gamma S_K} + \frac{2 B_0}{\eta S_K} + \left(\frac{16\gamma}{1 - \sqrt{1-\alpha}} + \frac{8\gamma}{\eta}\right) \left(L_0 + 4 L_1^2 \delta^{\mathrm{inf}}\right) \\
& + \left(\frac{4\eta\sqrt{1-\alpha}}{1 - \sqrt{1-\alpha}} + \frac{2\sqrt{\eta}}{\sqrt{n}}\right) \sigma,
\end{aligned}
\tag{22}
$$

where in the last inequality we use $V_{K+1} \geq 0$ and $\beta_{-1} = 1$. Next, we estimate $S_K$:

$$
\begin{aligned}
S_K \quad = \quad & \sum_{k=0}^{K} \beta_k \geq (K+1)\beta_K = \frac{K+1}{\left(1 + \frac{64 L_1^2 \gamma^2}{1 - \sqrt{1-\alpha}} + \frac{32 L_1^2 \gamma^2}{\eta}\right)^{K+1}} \\
\geq \quad & \frac{K+1}{\exp\left(\frac{64 L_1^2 \gamma^2 (K+1)}{1 - \sqrt{1-\alpha}} + \frac{32 L_1^2 \gamma^2 (K+1)}{\eta}\right)}.
\end{aligned}
\tag{23}
$$

Since $\eta = \frac{1}{(K+1)^{1/2}}$ and $\gamma = \frac{\gamma_0}{(K+1)^{3/4}}$ with $\gamma_0 \leq \frac{1}{16 L_1} \min\left\{(K+1)^{1/2}(1 - \sqrt{1-\alpha}), 1\right\}$, we have $\frac{32 L_1^2 \gamma^2 (K+1)}{\eta} \leq \frac{1}{4}$ and $\frac{64 L_1^2 \gamma^2 (K+1)}{1 - \sqrt{1-\alpha}} \leq \frac{1}{4}$. Plugging these inequalities into (23), we get $S_K \geq (K+1)/\exp(1/2) \geq (K+1)/2$. Using this lower bound for $S_K$ and $\eta = \frac{1}{(K+1)^{1/2}}, \gamma = \frac{\gamma_0}{(K+1)^{3/4}}$ in

 , we get

$$\min_{k=0,\ldots,K} \mathrm{E}\left[\left\|\nabla f(x^k)\right\|\right] \leq \frac{2V_0}{\gamma_0(K+1)^{1/4}} + \frac{4B_0}{(K+1)^{1/2}}$$
$$+ \left(\frac{16\gamma_0}{(1-\sqrt{1-\alpha})(K+1)^{3/4}} + \frac{8\gamma_0}{(K+1)^{1/4}}\right)\left(L_0 + 4L_1^2\delta^{\mathrm{inf}}\right)$$
$$+ \left(\frac{4\sqrt{1-\alpha}}{(1-\sqrt{1-\alpha})(K+1)^{1/2}} + \frac{2}{\sqrt{n}(K+1)^{1/4}}\right)\sigma.$$

For the convenience, we define $C_\alpha := 1 - \sqrt{1-\alpha}$. Then, by definition of $V_0$, we have

$$\frac{2V_0}{\gamma_0(K+1)^{1/4}} = \frac{2\delta^0}{\gamma_0(K+1)^{1/4}} + \frac{2A_0}{C_\alpha(K+1)} + \frac{2(1-C_\alpha)C_0}{C_\alpha(K+1)}.$$

Moreover, since $g_i^{-1} = 0$ and $v_i^{-1} = \nabla f_i(x_i^0; \xi_i^0)$ for all $i = 1, \ldots, n$ with independent $\{\xi_i^0\}_{i=1}^n$, we have $v_i^0 = \nabla f_i(x_i^0; \xi_i^0)$ and $g_i^0 = \mathcal{C}^0(\nabla f_i(x_i^0; \xi_i^0))$ for all $i = 1, \ldots, n$ and

$$A_0 = \frac{1}{n}\sum_{i=1}^n \mathrm{E}\left[\left\|\nabla f_i(x_i^0; \xi_i^0) - \mathcal{C}^0(\nabla f_i(x_i^0; \xi_i^0))\right\|\right]$$
$$\overset{(3)}{\leq} \frac{\sqrt{1-\alpha}}{n}\sum_{i=1}^n \mathrm{E}\left[\left\|\nabla f_i(x_i^0; \xi_i^0) - \nabla f_i(x_i^0)\right\|\right] \overset{(4)}{\leq} (1-C_\alpha)\sigma,$$
$$C_0 = \frac{1}{n}\sum_{i=1}^n \mathrm{E}\left[\left\|\nabla f_i(x_i^0; \xi_i^0) - \nabla f_i(x_i^0)\right\|\right] \overset{(4)}{\leq} \sigma,$$
$$B_0 = \mathrm{E}\left[\left\|\frac{1}{n}\sum_{i=1}^n \left(\nabla f_i(x_i^0; \xi_i^0) - \nabla f_i(x_i^0)\right)\right\|\right]$$
$$= \sqrt{\frac{1}{n^2}\sum_{i=1}^n \mathrm{E}\left[\left\|\nabla f_i(x_i^0; \xi_i^0) - \nabla f_i(x_i^0)\right\|^2\right]} \overset{(4)}{\leq} \frac{\sigma}{\sqrt{n}}.$$

Using these inequalities, we get

$$\min_{k=0,\ldots,K} \mathrm{E}\left[\left\|\nabla f(x^k)\right\|\right] \leq \frac{2\delta^0}{\gamma_0(K+1)^{1/4}} + \frac{2A_0}{C_\alpha(K+1)} + \frac{2(1-C_\alpha)C_0}{C_\alpha(K+1)} + \frac{4B_0}{(K+1)^{1/2}}$$
$$+ \left(\frac{16\gamma_0}{C_\alpha(K+1)^{3/4}} + \frac{8\gamma_0}{(K+1)^{1/4}}\right)\left(L_0 + 4L_1^2\delta^{\mathrm{inf}}\right)$$
$$+ \frac{4(1-C_\alpha)\sigma}{C_\alpha(K+1)^{1/2}} + \frac{2\sigma}{\sqrt{n}(K+1)^{1/4}}$$
$$\leq \frac{2\delta^0}{\gamma_0(K+1)^{1/4}} + \frac{4(1-C_\alpha)\sigma}{C_\alpha(K+1)} + \frac{4\sigma}{\sqrt{n}(K+1)^{1/2}}$$
$$+ \left(\frac{16\gamma_0}{C_\alpha(K+1)^{3/4}} + \frac{8\gamma_0}{(K+1)^{1/4}}\right)\left(L_0 + 4L_1^2\delta^{\mathrm{inf}}\right)$$
$$+ \frac{4(1-C_\alpha)\sigma}{C_\alpha(K+1)^{1/2}} + \frac{2\sigma}{\sqrt{n}(K+1)^{1/4}}$$
$$\leq \frac{2\delta^0}{\gamma_0(K+1)^{1/4}} + \left(\frac{16\gamma_0}{C_\alpha(K+1)^{3/4}} + \frac{8\gamma_0}{(K+1)^{1/4}}\right)\left(L_0 + 4L_1^2\delta^{\mathrm{inf}}\right)$$
$$+ \frac{8(1-C_\alpha)\sigma}{C_\alpha(K+1)^{1/2}} + \frac{6\sigma}{\sqrt{n}(K+1)^{1/4}},$$

which concludes the proof since $\frac{1-C_\alpha}{C_\alpha} \leq \frac{2\sqrt{1-\alpha}}{\alpha}$ and $\frac{1}{C_\alpha} \leq \frac{1}{\alpha}$

# F   Extension to Strongly Convex and Convex Problems

Our current analysis for ||EF21|| and ||EF21-SGDM||, which are initially developed for minimizing non-convex functions, can be extended to strongly convex and convex functions.

**Strongly convex problems.**   We can extend the convergence for ||EF21|| and ||EF21-SGDM|| to minimize strongly convex functions. Applying the $\mu$-strong convexity condition of the function $f$, i.e. $\left\|\nabla f(x^k)\right\|^2 \geq 2\mu(f(x^k) - f(x^\star))$, where $x^\star = \arg\min_{x \in \mathbb{R}^d} f(x)$, into the convergence bounds in Theorems 1 and 2 yields the convergence results in $\min_{k=0,1,\ldots,K} \mathrm{E}\left[\sqrt{f(x^k) - f(x^\star)}\right]$. However, these results do not imply the standard exponential convergence typically expected in strongly convex problems. This theoretical gap suggests a need for new analytical techniques, which involves tighter Lyapunov functions or more refined descent inequalities tailored to strongly convex functions.

**Convex problems.**   We can extend the convergence for minimizing convex functions. This can be achieved by assuming that there exists the iterates $\{x^k\}$ satisfying $\left\|x^k - x^\star\right\| \leq R$ for some $R > 0$. Hence, the convexity of the function $f$ implies that

$$f(x^k) - f(x^\star) \leq \left\|\nabla f(x^k)\right\| \left\|x^k - x^\star\right\| \leq R \left\|\nabla f(x^k)\right\|.$$

Applying the above inequality to Theorems 1 and 2 yields the convergence bounds in $\min_{k=0,1,\ldots,K} \mathrm{E}\left[f(x^k) - f(x^\star)\right]$.

# G  Additional Experimental Results

In this section, we provide additional results for minimizing nonconvex polynomial functions, and for training the ResNet-20 model over the CIFAR-10 dataset.

## G.1  Minimization of Nonconvex Polynomial Functions

We ran ||EF21|| and EF21 in a single-node setting ($n = 1$) for solving the following problem:

$$\min_{x \in \mathbb{R}^d} \left\{ f(x) := \underbrace{\sum_{i=1}^{d} a_i x_i^4}_{=:g(x)} + \lambda \underbrace{\sum_{i=1}^{d} \frac{x_i^2}{1 + x_i^2}}_{=:h(x)} \right\}, \tag{24}$$

where $a_i > 0$, $i = 1, \ldots, d$, $\lambda > 0$.

Let us show that $f(x)$ is non-convex (for the specific choice of $a_i$) and $(L_0, L_1)$-smooth. First, we prove that $f(x)$ is non-convex. Indeed,

$$\nabla^2 f(x) = \nabla^2 g(x) + \nabla^2 h(x)$$

$$= 12 \, \mathrm{diag} \left\{ a_1 x_1^2, \ldots, a_d x_d^2 \right\} + 2\lambda \, \mathrm{diag} \left\{ \frac{1 - 3x_1^2}{(1 + x_1^2)^3}, \ldots, \frac{1 - 3x_d^2}{(1 + x_d^2)^3} \right\},$$

is not positive definite matrix if we choose $a_i = \frac{\lambda}{24}$, $x_i = \pm 1$ for $i = 1, \ldots, d$.

Second, we find $L_0, L_1 > 0$ such that

$$\left\| \nabla^2 f(x) \right\| \leq L_0 + L_1 \left\| \nabla f(x) \right\|, \quad \forall x \in \mathbb{R}^d.$$

This condition is equivalent to Assumption 3 (generalized smoothness) with $L_0, L_1$ [16, Theorem 1]. Let us fix some $L_1 > 0$ and choose $L_0 = \frac{9\lambda d^2}{2L_1^2} + 2\lambda$. Since $\nabla^2 h(x) \preccurlyeq 2\lambda I$,

$$\left\| \nabla^2 f(x) \right\| = \left\| \nabla^2 g(x) + \nabla^2 h(x) \right\| \leq \left\| \nabla^2 g(x) \right\| + \left\| \nabla^2 h(x) \right\|$$

$$\leq 12 \sqrt{a_1^2 x_1^4 + \ldots + a_d^2 x_d^4} + 2\lambda$$

$$\leq 12 \left( a_1 x_1^2 + \ldots + a_d x_d^2 \right) + 2\lambda.$$

Also, notice that

$$\|\nabla f(x)\| = \|\nabla g(x) + \nabla h(x)\| = \sqrt{\left( 4a_1 x_1^2 + \frac{2\lambda}{(1 + x_1^2)^2} \right)^2 x_1^2 + \ldots + \left( 4a_d x_d^2 + \frac{2\lambda}{(1 + x_d^2)^2} \right)^2 x_d^2}$$

$$\geq 4 \sqrt{a_1^2 x_1^6 + \ldots + a_d^2 x_d^6}$$

$$\overset{(*)}{\geq} \frac{4}{\sqrt{d}} \left( a_1 |x_1|^3 + \ldots + a_d |x_d|^3 \right),$$

where (*) results from the fact that $\|x\|_1 \leq \sqrt{d} \, \|x\|$ for $x \in \mathbb{R}^d$. Our goal is to show that

$$12 \left( a_1 x_1^2 + \ldots + a_d x_d^2 \right) \leq \tilde{L}_0 + \frac{4L_1}{\sqrt{d}} \left( a_1 |x_1|^3 + \ldots + a_d |x_d|^3 \right), \quad \tilde{L}_0 = L_0 - 2\lambda.$$

To show this, we consider two cases: if $|x_i| \leq \frac{3\sqrt{d}}{L_1}$, and otherwise.

1. If $|x_i| \leq \frac{3\sqrt{d}}{L_1}$ for all $i = 1, \ldots, d$, then $12 a_i x_i^2 \leq \frac{108 a_i d}{L_1^2}$. Thus, $12 \left( a_1 x_1^2 + \ldots + a_d x_d^2 \right) \leq \frac{108 \lambda d^2}{24 L_1^2} = \tilde{L}_0$.

2. If $|x_j| > \frac{3\sqrt{d}}{L_1}$ for some $j = 1, \ldots, d$, then $12 a_j x_j^2 < \frac{4L_1}{\sqrt{d}} a_j |x_j|^3$, and the sum of the remaining terms (such that $|x_i| \leq \frac{3\sqrt{d}}{L_1}$) in $12 \left( a_1 x_1^2 + \ldots + a_d x_d^2 \right)$ can be upper bounded by $\tilde{L}_0$.

In conclusion, $f(x)$ is $(L_0, L_1)$-smooth, where $L_1$ is any positive constant and $L_0 = \frac{9\lambda d^2}{2L_1^2} + 2\lambda$.

Additionally, we can show that under certain additional constraints, $f(x)$ is $L$-smooth with $L = \frac{\lambda\sqrt{d}D^2}{2} + 2\lambda$. If $|x_i| \leq D$ for all $i = 1, \ldots, d$, then

$$\left\|\nabla^2 f(x)\right\| \leq 12\sqrt{a_1^2 x_1^4 + \ldots + a_d^2 x_d^4} + 2\lambda \leq \frac{\lambda\sqrt{d}D^2}{2} + 2\lambda = L,$$

In the experiments, we estimate $D$ based on the initial point $x^0 \in \mathbb{R}^d$.

In the following experiments, we used a top-$k$ sparsifier with $k = 1$ and $\alpha = k/d$, setting $d = 4$, $L_1 = \{1, 4, 8\}$, and $L_0 = 4$ (adjusting $\lambda$ to maintain a constant $L_0$). The initial values $x^0$ were drawn from a normal distribution, $x_i^0 \sim \mathcal{N}(20, 1)$ for $i = 1, \ldots, d$, with $D$ estimated as 20. For EF21, we set $\gamma_k = \frac{1}{L + L\sqrt{\frac{\beta}{\theta}}}$, using $\theta = 1 - \sqrt{1-\alpha}$ and $\beta = \frac{1-\alpha}{1-\sqrt{1-\alpha}}$, according to Theorem 1 of [8]. For $\|\text{EF21}\|$, we chose $\gamma_k = \frac{1}{2c_1}$ with $c_1 = \frac{L_1}{2} + 2\frac{\sqrt{1-\alpha}L_1}{1-\sqrt{1-\alpha}}$ from Theorem 3, and $\gamma_k = \frac{\gamma_0}{\sqrt{K+1}}$ with $\gamma_0 > 0$, as specified in Theorem 1 with $n = 1$.

**The impact of $\gamma_0$ and $K$ on the convergence of $\|\text{EF21}\|$.** First, we investigate the impact of $\gamma_0$ and $K$ on the convergence of $\|\text{EF21}\|$. We evaluated $\gamma_0$ from the set $\{0.1, 1, 10\}$, and plotted the histogram representing the number of iterations required to achieve the target accuracy of $\|\nabla f(x)\|^2 < \epsilon$ with $\epsilon = 10^{-4}$, using the stepsize rule $\gamma = \frac{\gamma_0}{\sqrt{K+1}}$. For each $\gamma_0$, we determined $K$ as the minimum number of iterations required to achieve the desired accuracy, found through a grid search with step sizes of 500 for $\gamma_0 = 1, 10$ and 5000 for $\gamma_0 = 0.1$. From Figure 4, for small values of $\gamma_0$, such as

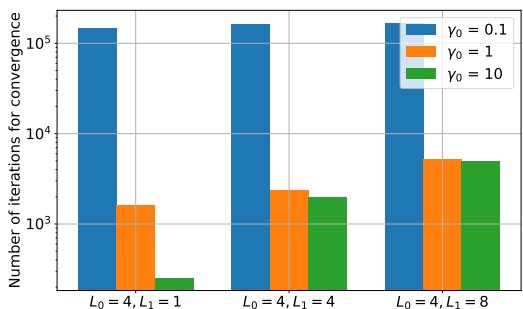

Figure 4: Number of iterations required to achieve the desired accuracy, $\|\nabla f(x)\|^2 < \epsilon$, $\epsilon = 10^{-4}$, using $\|\text{EF21}\|$ with $\gamma = \frac{\gamma_0}{\sqrt{K+1}}$ for different values of $L_0$ and $L_1$.

0.1, significantly more iterations are required to reach convergence compared to $\gamma_0$ values of 1 and 10, which show similar performance (with the exception of the $L_0 = 4$, $L_1 = 1$ case, where $\gamma_0 = 10$ converges faster). Based on this observation, we use $\gamma_0 = 1$ in all subsequent experiments and adjust only $K$ to achieve convergence, identifying the minimum number of iterations needed to reach the target accuracy through a grid search with a step size of 500.

**Comparisons between EF21 and $\|\text{EF21}\|$.** Next, we evaluate the performance of EF21 and $\|\text{EF21}\|$ for a fixed $L_0 = 4$ and varying $L_1$ values of $\{1, 4, 8\}$. From Figure 1, $\|\text{EF21}\|$, regardless of the chosen stepsize $\gamma$, achieves the desired accuracy $\|\nabla f(x)\|^2 < \epsilon$ with $\epsilon = 10^{-4}$ faster than EF21. Initially, however, EF21 converges more quickly, likely because $\|\text{EF21}\|$ employs normalized gradients, which can be slower at the start due to the large gradients when the initial point is far from the stationary point. Moreover, as $L_1$ increases, both methods show slower convergence.

## G.2 ResNet20 Training over CIFAR-10

We included additional experimental results from running EF21 and $\|\text{EF21}\|$ for training the ResNet20 model over the CIFAR-10 dataset. The parameter details were set to be the same as those in Section 6.2, with the exception that we vary $k = 0.01d, 0.5d$ for a top-$k$ sparsifier. From Figures 5 and 6, $\|\text{EF21}\|$ attains a higher accuracy improvement than EF21, across different sparsification levels $k$.

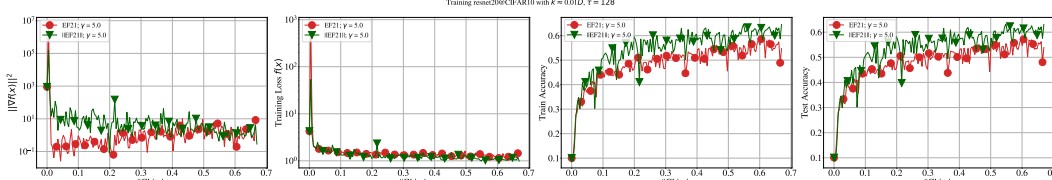

Figure 5: ResNet20 training on CIFAR-10 by using EF21 and ||EF21|| under the same stepsize $\gamma = 5$ and $k = 0.01d$ for a top-$k$ sparsifier.

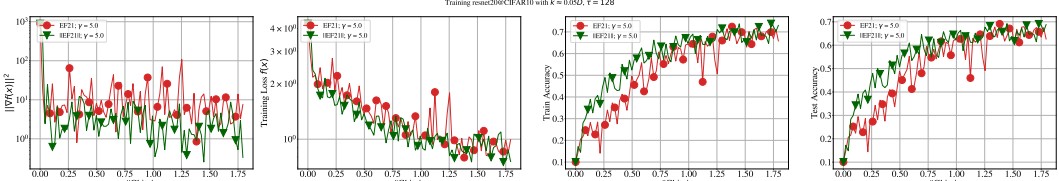

Figure 6: ResNet20 training on CIFAR-10 by using EF21 and ||EF21|| under the same stepsize $\gamma = 5$ and $k = 0.05d$ for a top-$k$ sparsifier.

# H   Omitted Proof for Smoothness Parameters of Logistic Regression

In this section, we prove the generalized smoothness parameters $L_0, L_1$ for logistic regression problems with a nonconvex regularizer, which are the following problems

$$\min_{x\in\mathbb{R}^d}\left\{f(x):=\frac{1}{n}\sum_{i=1}^n f_i(x):=\frac{1}{n}\sum_{i=1}^n \underbrace{\log(1+\exp(-b_i a_i^T x))}_{=:\tilde{f}_i(x)} + \lambda\underbrace{\sum_{j=1}^d \frac{x_j^2}{1+x_j^2}}_{=:h(x)}\right\},$$

where $a_i \in \mathbb{R}^d$ is the $i^{\text{th}}$ feature vector of matrix $A$ with its class label $b_i \in \{-1, 1\}$, $\lambda > 0$.

First, we can prove that $f(x)$ is $L$-smooth with $L = \frac{1}{4n}\|A\|^2 + 2\lambda$, and that each $f_i(x)$ is $\hat{L}_i$-smooth with $\hat{L}_i = \frac{1}{4}\|a_i\|^2 + 2\lambda$.

Next, we show that each $f_i(x)$ is generalized smooth with $L_0 = 2\lambda + \lambda\sqrt{d}\max_i \|a_i\|$ and $L_1 = \max_i \|a_i\|$, when the Hessian exists. By the fact that

$$\nabla\tilde{f}_i(x) = -\frac{\exp(-b_i a_i^T x)}{1+\exp(-b_i a_i^T x)}b_i a_i, \quad \text{and} \quad \nabla^2\tilde{f}_i(x) = \frac{\exp(-b_i a_i^T x)}{(1+\exp(-b_i a_i^T x))^2}b_i^2 a_i a_i^T,$$

we have

$$
\begin{aligned}
\left\|\nabla^2\tilde{f}_i(x)\right\| &\overset{b_i\in\{-1,1\}}{=} \frac{\exp(-b_i a_i^T x)}{(1+\exp(-b_i a_i^T x))^2}\lambda_{\max}(a_i a_i^T)\\
&= \frac{\exp(-b_i a_i^T x)}{(1+\exp(-b_i a_i^T x))^2}\|a_i\|^2\\
&= \frac{\|a_i\|}{1+\exp(-b_i a_i^T x)}\left\|\nabla\tilde{f}_i(x)\right\|\\
&\leq \|a_i\|\left\|\nabla\tilde{f}_i(x)\right\|.
\end{aligned}
\tag{25}
$$

After adding the nonconvex regularizer $h(x)$, we can show the following inequalities:

$$
\begin{aligned}
\left\|\nabla^2 f_i(x)\right\| &\leq \left\|\nabla^2\tilde{f}_i(x)\right\| + \left\|\nabla^2 h(x)\right\|\\
&\leq \left\|\nabla^2\tilde{f}_i(x)\right\| + 2\lambda,
\end{aligned}
\tag{26}
$$

and

$$
\begin{aligned}
\|\nabla f_i(x)\| \geq \left\|\nabla\tilde{f}_i(x)\right\| - \|\nabla h(x)\| &= \left\|\nabla\tilde{f}_i(x)\right\| - \sqrt{\left(\frac{2\lambda x_1}{(1+x_1^2)^2}\right)^2 + \ldots + \left(\frac{2\lambda x_d}{(1+x_d^2)^2}\right)^2}\\
&\geq \left\|\nabla\tilde{f}_i(x)\right\| - \sqrt{\lambda^2 + \ldots + \lambda^2}\\
&= \left\|\nabla\tilde{f}_i(x)\right\| - \lambda\sqrt{d}.
\end{aligned}
\tag{27}
$$

By combining inequalities (25), (26), and (27), we obtain

$$
\begin{aligned}
\left\|\nabla^2 f_i(x)\right\| &\leq \left\|\nabla^2\tilde{f}_i(x)\right\| + 2\lambda\\
&\leq \|a_i\|\left\|\nabla\tilde{f}_i(x)\right\| + 2\lambda\\
&\leq 2\lambda + \lambda\sqrt{d}\|a_i\| + \|a_i\|\|\nabla f_i(x)\|.
\end{aligned}
$$

In conclusion, $\left\|\nabla^2 f_i(x)\right\| \leq L_0 + L_1\|\nabla f_i(x)\|$ with $L_0 \leq 2\lambda + \lambda\sqrt{d}\|a_i\|$, and $L_1 \leq \|a_i\|$. This condition is equivalent to Assumption 3 (generalized smoothness) with $L_0, L_1$ [16, Theorem 1].

