# OpenReview forum: "Error Feedback under $(L_0,L_1)$-Smoothness: Normalization and Momentum"
_NeurIPS.cc/2025/Conference — NeurIPS 2025 poster_

### Official Review · Reviewer_Xvhq · 2025-06-29

**Clarity:** 3
**Significance:** 3
**Originality:** 3
**Rating:** 5
**Confidence:** 2

**Summary:**

This paper addresses the challenge of training deep neural networks with error feedback algorithms under generalized smoothness assumptions, specifically $(L_0,L_2)$-smoothness, which is more representative of modern machine learning problems than traditional smoothness assumptions.

**Questions:**

- **Anonymous GitHub Link:** As of June 29, I attempted to access the anonymous GitHub repository, but the link has expired.
- **Proof:** I did not follow every proof detail; however, the overall results appear reasonable and reliable.
- **Experimental Questions:**
  1. **Logistic Regression:** The proposed method substantially outperforms EF21. Could you specify the exact step sizes used for both algorithms?
  2. **Related Work Comparison:** You identify [25] and [26] as the most relevant works. Maybe, you can include a direct comparison of your experimental results with the findings in these references.
  3. **Machine number:** I think the $n$ become the number of data volume in experiment section. Please point out how many machines (nodes) are you used in the experiments.

**Ethical Concerns:**

["NO or VERY MINOR ethics concerns only"]

**Final Justification:**

I must acknowledge that I did not grasp every detail of the manuscript; however, overall, its logic is relatively clear and its contribution is well-defined.

**Quality:**

3

**Strengths And Weaknesses:**

- The paper provides the proof of convergence for normalized error feedback algorithms in $(L_0,L_1)$ generated smoothness. These analyses apply to distributed settings with data heterogeneity conditions.

- The primary limitation is that the current analysis relies on constant stepsizes (depends on the iterations budget $K$), hence it is not a any-time optimal algorithm.

---

> ### Author Rebuttal · Authors · 2025-07-30
>
> We thank the reviewer for their constructive feedback and for recognizing the relevance of our generalized smoothness analysis and its applicability to modern distributed training scenarios.
>
>
> > **Q1**: The primary limitation is that the current analysis relies on constant stepsizes (depends on the iterations budget ), hence it is not a any-time optimal algorithm.
>
>
> **A1**: We appreciate this important observation. As you pointed out, our current theoretical analysis assumes a constant stepsize that depends on the total number of iterations, which limits the any-time nature of the algorithm.
> We believe that our current results can be extended to the case of decreasing stepsizes, which would remove the dependency on knowing the total number of iterations. For example, in the case of || EF21 || under generalized/relaxed smoothness assumptions, we can show that by using a decreasing stepsize on the form
> $$
> \gamma_k = \gamma_0/(k+1)^{1/2+\nu} \quad \text{for} \quad \nu \in (0,1/2),
> $$
> yields a convergence rate of
> $$
> \min_{k\in[0,K]} \mathbb{E}|| \nabla f(x_k)||  = \mathcal{O}\left( \frac{1}{(K+1)^{1/2-\nu}} \right).
> $$
> We prove this result from the descent inequality we obtained for  || EF21 || under generic decreasing stepsizes (Theorem 1):
> $$
> V_{k+1} \leq (1+A \gamma_k^2 \exp(L_1\gamma_k)) V_k - \gamma_k W_k + B \exp(L_1\gamma_k) \gamma_k^2,
> $$
> where $V_k = \mathbb{E}{f(x_k)-f_{\inf}}$ and $W_k = \mathbb{E}\| \nabla f(x_k)\|^2$.
> Since $\gamma_k \leq \gamma_0$, this inequality simplifies to:
> $$
> V_{k+1} \leq (1+ \tilde A \gamma_k^2) V_k - \gamma_k W_k + \tilde B \gamma_k^2,
> $$
> where $\tilde A = A \exp(L_1\gamma_0)$ and $\tilde B = B \exp(L_1\gamma_0)$.
>
> Finally, defining a sequence $\alpha_k = \alpha_{k-1}/(1+\tilde A \gamma_k^2)$, applying the fact that  $\min_{k \in [0,K]} W_k \leq \frac{1}{\sum_{k=0}^K \alpha_k\gamma_k} \sum_{k=0}^K \alpha_k \gamma_k W_k$, and using the above inequality, we obtain the final convergence result stated above.
>
> Furthermore, we believe that similar convergence results can be established for  || EF21-SGDM || under the decreasing stepsize, e.g. by choosing $\gamma_k=\gamma_0/(k+1)^{3/4}$ and $\eta_k = 1/(k+1)^{1/2}$. However, this requires us to revisit key lemmas related to the decreasing stepsizes, as presented, e.g., by Hübler et al. (2024).
>
>
> > **Q2**: Logistic Regression: The proposed method substantially outperforms EF21. Could you specify the exact step sizes used for both algorithms?
>
>
> **A2**: We benchmarked normalized EF21 and original EF21 algorithms **with their own theoretical stepsizes** for solving logistic regression problems. The stepsize of normalized EF21 is $\gamma = 1/\sqrt{K+1}$ by Theorem 1 of our paper, while that of original EF21 is $\gamma = 1/(L + \tilde L \sqrt{\beta/\theta})$ with $\tilde L = \sqrt{\sum_i L_i^2/n}$ and $\theta,\beta$ given by Theorem 1 of [8]. Here, $L,\tilde L$ is estimated from the loss function, while $\theta,\beta$ is tuned according to $\alpha = k/d$ for the top-k sparsifier and for each $d$-dimensional feature vector of the data matrix.
>
>
> > **Q3**: Related Work Comparison: You identify [25] and [26] as the most relevant works. Maybe, you can include a direct comparison of your experimental results with the findings in these references.
>
>
> **A3**: We include [25], because clipping and normalization are two popular operations for stabilizing optimization algorithms to minimize relaxed smooth functions. As clipping is the generalization of normalization, we agree that it is interesting to extend our current theorems for the cases when we apply clipping rather than normalization, and to benchmark our modified algorithms empirically.
> Regarding [26], we reference it (Lines 141–146) because it also addresses convergence under relaxed smoothness assumptions. However, their approach relies on additional restrictive conditions—such as almost sure variance bounds and symmetric noise distributions—which our analysis avoids. Moreover, we do not benchmark against their algorithms, as their methods do not incorporate error feedback, a key feature of our proposed approach.
>
>
> > **Q4**: Machine number: I think the $n$  become the number of data volume in experiment section. Please point out how many machines (nodes) are you used in the experiments.
>
>
> **A4**: Thank you for pointing this out. You are correct – there is an overload of notation. In our experiments on nonconvex polynomial minimization and logistic regression, we used $n$ to denote the number of data points. In contrast, for the ResNet-20 training on CIFAR-10, $n$ refers to the number of clients (machines), which we set to $n=5$. We will revise this part of the paper to clearly distinguish between the number of data points and the number of clients, and adopt consistent, unambiguous notation throughout.

---

> > ### Comment · Reviewer_Xvhq · 2025-08-01
> > **Reply to rebuttal**
> >
> > I mostly agree with the point of view presented in the rebuttal, and I will raise the score accordingly.

---

> > > ### Author Response · Authors · 2025-08-04
> > >
> > > > I mostly agree with the point of view presented in the rebuttal, and I will raise the score accordingly.
> > >
> > > We are glad that our clarifications addressed most of your concerns. Thank you for acknowledging our contributions and for raising your score for our paper.

---

### Official Review · Reviewer_nR5L · 2025-06-30

**Clarity:** 2
**Significance:** 3
**Originality:** 3
**Rating:** 5
**Confidence:** 3

**Summary:**

In this work the authors studied the EF21 algorithm with normalisation and the L0-L1 smoothness assumption. The authors analyzed the normalised EF21 algorithm and normalised EF21-SGDM algorithm theoretically, and conducted experiments evaluating the performance of normalised EF21 against the ordinary EF21.

**Questions:**

1. Can the authors provide clearer explanations regarding their proof techniques? What are the choices of the parameters in the Lyapunov functions? How are these choices derived and why are they critical for the analysis of the normalised method (in contrast to the choices in the EF21 paper)? There are also some recent papers that got rid of the need of Lyapunov functions altogether, by upper bounding the sum of the error terms together [1]. Such an approach eliminated the need for guessing the parameters in the Lyapunov function. Since the authors believe that the novel technical contribution of the proof lies in the construction of the Lyapunov function, I wonder if it's possible to follow the approach in [1] and obtain an analysis of the normalised EF21 without the Lyapunov function?

2. Can the authors add a discussion regarding Assumption 2? In particular, it would be interesting to understand why is this assumption necessary? Would the analysis of EF21 be improved when given this assumption? This should also be complimented by a discussion on how would $\delta^{\inf}$ affect the comparisons of convergence rates

3. Can the authors please also compare the normalised EF21 with other popular compressed optimization methods, including the original EF, and maybe also EControl or some other variants of EF?



[1] Gao et. al., Accelerated Distributed Optimization with Compression and Error Feedback, 2025

**Ethical Concerns:**

["NO or VERY MINOR ethics concerns only"]

**Final Justification:**

The authors have addressed my questions and I have raised the score. I strongly encourage the authors to include their discussions in the rebuttal in the main text of the paper.

**Limitations:**

see weakness and questions.

I think the results of the paper are interesting and I'll be happy to raise the score if my questions can be addressed.

**Quality:**

2

**Strengths And Weaknesses:**

I think from a theory perspective this is an interesting paper. No EF21-style methods, or indeed, no EF methods, has been analysed with normalised stepsizes and L0-L1 smoothness prior to this work. So the results are new and I think the techniques employed here would be of interests to other people working on the same topics as well. In particular, controlling the errors with varying stepsizes for EF, or EF21 style methods, is non-trivial (unless when the stepsizes are non-increasing, which is not the case with normalised stepsizes). However, in its current form, the paper still has many weaknesses, and I discuss these below:

1. Lack of clear discussion on the technical innovations in the proofs: the authors had a brief discussion of their proof techniques at the end of Section 5, but this discussion, to put it mildly, is non-informative. The authors claimed that the key difference is a new choice of Lyapunov function. But it's unclear why is that the case given the discussion. It seems that the only difference is just with the choice of parameter A and B. The authors didn't discuss the choices of A and B, so for all we know they might as well be the same.

2. Stronger assumption: The authors claimed that Assumption 2 is standard for analysing optimisation algorithms. I find that unconvincing. In particular, as far as I'm aware, most existing works on EF and EF21 only requires Assumption 1, instead of Assumption 2. Here it seems that Assumption 2 acts as some sort of relaxation of the overparametrization assumption, which would typically be considered a very strong assumption. I'm not against introducing stronger assumptions when needed, but this should be adequately discussed.

3. The experiments section only compares the proposed algorithm against EF21. This seems a bit insufficient to me. There are many other state of the arts algorithms in compressed optimization methods, and I think it's hard to gauge the actual practical value of the proposed algorithm if it's only compared against EF21.

---

> ### Author Rebuttal · Authors · 2025-07-30
>
> We sincerely thank the reviewer for their thoughtful feedback and for acknowledging the novelty of our theoretical techniques and the potential impact of our work. We address each concern below:
>
> > **Q1**: Can the authors provide clearer explanations regarding their proof techniques? What are the choices of the parameters in the Lyapunov functions? How are these choices derived and why are they critical for the analysis of the normalised method (in contrast to the choices in the EF21 paper)? There are also some recent papers that got rid of the need of Lyapunov functions altogether, by upper bounding the sum of the error terms together [1]. Such an approach eliminated the need for guessing the parameters in the Lyapunov function. Since the authors believe that the novel technical contribution of the proof lies in the construction of the Lyapunov function, I wonder if it's possible to follow the approach in [1] and obtain an analysis of the normalised EF21 without the Lyapunov function?
>
> **A1**: Thank you for the thoughtful questions. We address them below in the following points:
>
> **Clarification on proof techniques and Lyapunov function parameters:** The structure of our Lyapunov function differs from EF21 primarily in the second term: we use $|| v^k_i -\nabla f(x^k) ||$, while EF21 uses $|| v^k_i-\nabla f(x^k) ||^2$. This change is driven by the nature of the descent lemma in our analysis. Due to normalization in our method, we need to control the quantity $|| g^k -\nabla f(x^k) ||$, whereas EF21 deals with $|| g^k -\nabla f(x^k) ||^2$. This distinction affects the entire structure of our Lyapunov function.
>
> Additionally, our analysis operates under a **generalized smoothness condition**, unlike EF21, which assumes standard smoothness. The generalized smoothness makes the analysis more intricate, particularly in deriving recursive inequalities. This is most evident in Lemma 4, which forms the core of our recursive argument and requires significantly different handling compared to EF21.
>
> Moreover, the proof technique for normalized EF21-SGDM deviates even more significantly from prior works. Specifically, we introduce a novel Lyapunov function that differs from the one used by Fatkhullin et al. [10] (see our submission), derive an upper bound for $\mathbb{E}[\|\| v^k - \nabla f(x^k) \|\|]$ via a new route (see Lemma 9), and employ the Lyapunov function in a fundamentally different manner.
> In particular, after establishing the descent inequality in the proof of Theorem 2 (see the inequality following Line 624), we face a key challenge: the right-hand side involves non-telescoping terms, namely $\left(1 + \frac{64L_1^2 \gamma^2}{1 - \sqrt{1-\alpha}}\right)V_k + 32L_1^2\gamma^2 \sum_{t=0}^{k-1} (1-\eta)^{k-t} V_t$ which prevents a straightforward summation as done in the analyses of EF21 or EF21-SGDM.
>
> To overcome this, we introduce carefully chosen weights $\beta_k$ (defined in Line 625) and apply a weighted summation strategy. This allows us to effectively upper-bound the double-sum term on the right-hand side and complete the proof of convergence (see Lines 626–639).
>
> **On the choice of Lyapunov function parameters:** The parameters in our Lyapunov function are not guessed heuristically; instead, they are **systematically derived** to ensure the monotonic decrease of the Lyapunov function, i.e., to guarantee that $V_{k+1} \leq V_k$. This is crucial for proving convergence. We refer you to **Lemma 6**, where this parameter selection arises naturally from the analysis. The choice ensures that error terms are properly balanced across iterations to obtain a descent inequality.
>
> **On Lyapunov-free techniques and comparison with [1]:** We agree that the Lyapunov-free approach in [1]—which upper bounds the accumulated error without relying on a crafted Lyapunov function—is an elegant and promising alternative. However, to the best of our knowledge, we believe that **preferences between Lyapunov-style and Lyapunov-free analysis can be subjective.**
>
> Furthermore, the effectiveness of the Lyapunov-free approach is tied to their use of the **Similar Triangle Method**, which serves as a foundation for their new analysis. Therefore, applying a similar technique to our normalized method would be **non-trivial and may require significant modification** of both algorithmic structure and analytical tools. Additionally, **[1] focuses on convex and standard smooth settings, whereas our framework targets nonconvex problems with relaxed smoothness**, which poses different technical challenges.
> That said, exploring a Lyapunov-free analysis for our setting is an interesting direction for future work, and we thank you for raising this possibility.
>
>
> > **Q2**: Can the authors add a discussion regarding Assumption 2? In particular, it would be interesting to understand why is this assumption necessary? Would the analysis of EF21 be improved when given this assumption? This should also be complimented by a discussion on how would  affect the comparisons of convergence rates.
>
>
> **A2**: **Assumption 2 is necessary** to carry out our analysis under the **generalized smoothness condition**, which is more general than the standard $L$-smoothness used in EF21 and [1]. Specifically, this assumption **plays a crucial role in controlling the discrepancy between local and global function behavior**, which is essential for establishing descent bounds under our relaxed smoothness model. Notably, this assumption is made **in place of the commonly used uniformly bounded heterogeneity condition** found in existing works.
> Importantly, if we consider the special case where **$L_1 = 0$**, then our analysis simplifies considerably. In this case, the troublesome term involving the difference between $f^{\inf}$ and $f_i^{\inf}$ disappears, and **Assumption 2 becomes unnecessary**. This highlights that
>
>
> 1. The original EF21 and || EF21 || can operate under standard smoothness assumptions and do not require Assumption 2.
>
> 2. Assumption 2 is only invoked to handle the additional complexity introduced by **$(L_0,L_1)$-smoothness assumptions**.
>
> Furthermore, in terms of convergence rate in $\varepsilon$ under **$(L_0,L_1)$-smoothness assumptions**, our method, || EF21 ||, **matches** the EF21 rate. Furthermore, our convergence results are **parameter-agnostic**, meaning they do not require tuning based on problem-specific constants.
> However, the detailed comparison of constants in the rates (i.e., dependence on $L_0$, $L_1$, and $\delta^{\inf} := \frac{1}{n}\sum_{i=1}^n f^{\inf} - f_i^{\inf}$) is more nuanced due to the difference in assumptions and settings. If we set $L_1 = 0$, then the generalized smoothness reduces to standard smoothness, and our rate constants become comparable to those in EF21. In this case, we recover a similar rate up to minor differences in step size choices (e.g., $\gamma = \sqrt{\Delta / c_0}$ in our method).
> We are happy to include these discussions in the final version of our manuscript.
>
>
> > **Q3**: Can the authors please also compare the normalised EF21 with other popular compressed optimization methods, including the original EF, and maybe also EControl or some other variants of EF?
>
> **A3**: We already provide a direct empirical comparison between normalized EF21 and EF21 in Figure 1, which demonstrates the effectiveness of normalization to improve the convergence of EF21 algorithms to minimize the convex, relaxed smooth functions. Specifically, normalized EF21 allows for larger stepsizes and faster convergence than its non-normalized counterpart, original EF21. Our additional benchmarking results on logistic regression and ResNet-20 training further confirm these findings.
>
> Our main focus in this paper is to investigate **why normalization is important under relaxed smoothness assumptions**. While a direct comparison with other error feedback variants (e.g. **EControl**) may not directly address this core question, we agree that it would be valuable to explore how normalization interacts with these methods. Extending both the theoretical and empirical analysis to other error feedback variants could help determine whether normalization plays a similarly stabilizing or accelerating role in those contexts as well.

---

> > ### Comment · Reviewer_nR5L · 2025-08-01
> >
> > I would like to thank the reviewer for their detailed answers. I have raised my score.
> >
> > I would strongly encourage the authors to include in the main paper their discussions on the Lyapunov functions' construction, and Assumption 2. I believe these are very important part of the paper, and if you will, "the story", and should be presented clearly.
> >
> > Some other comments:
> >
> > - Lyapunov function: I thank the authors for the detailed explanations on the techniques used here. I want to clarify that I'm not trying to say that Lyapunov-free methods are better, but merely pointing it out as a possible alternative.
> >
> > - Assumption 2: it's interesting that the authors noted that Assumption 2 is some sort of a bounded heterogeneity assumption. This is a somewhat surprising feature of the generalised smoothness setting, considering that EF21 was introduced precisely to deal with heterogenous case. I'm curious if the authors believe that Assumption 2 might be removed with some other algorithmic tools?

---

> > > ### Author Response · Authors · 2025-08-04
> > >
> > > We are glad that our clarifications addressed your concerns. Thank you for acknowledging our contributions and for raising your score for our paper.
> > >
> > > Also, we appreciate your thoughtful feedback regarding the importance of including detailed discussions on the construction of Lyapunov functions and Assumption 2. We agree that these additions will enhance the quality of the paper and will include them into the final manuscript.
> > >
> > >
> > > > Lyapunov function: I thank the authors for the detailed explanations on the techniques used here. I want to clarify that I'm not trying to say that Lyapunov-free methods are better, but merely pointing it out as a possible alternative.
> > >
> > > Thank you for the clarification. We agree that Lyapunov-free methods are interesting alternatives for analyzing optimization methods. While our work builds on Lyapunov-based methods, exploring Lyapunov-free alternatives in our setting is a promising direction for future research.
> > >
> > >
> > > > Assumption 2: it's interesting that the authors noted that Assumption 2 is some sort of a bounded heterogeneity assumption. This is a somewhat surprising feature of the generalised smoothness setting, considering that EF21 was introduced precisely to deal with heterogenous case. I'm curious if the authors believe that Assumption 2 might be removed with some other algorithmic tools?
> > >
> > > Thank you for your insightful question. We agree that while EF21 does not require Assumption 2 to handle heterogeneity under traditional smoothness assumptions, our normalized EF21 method does require Assumption 2 under the relaxed smoothness setting. The key difference stems from the normalization step in normalized EF21, which introduces additional bias to the gradient beyond the compression error in the error feedback mechanism in EF21.
> > > Consequently, we believe that removing Assumption 2 entirely would likely require restructuring the normalized EF21 algorithm or incorporating variance-reduction techniques, such as momentum. We consider these promising directions for future work.

---

### Official Review · Reviewer_Acin · 2025-07-01

**Clarity:** 3
**Significance:** 3
**Originality:** 3
**Rating:** 5
**Confidence:** 5

**Summary:**

This paper explores the convergence properties of error feedback algorithms in distributed machine learning under generalized smoothness assumptions. Building on the existing EF21 error feedback algorithm, the authors introduce a new variant, **||EF21||**, which achieves \$O(1/\sqrt{K})\$ convergence for nonconvex problems without requiring data heterogeneity. Additionally, the paper extends these results to stochastic settings by incorporating momentum-based updates (**||EF21-SGDM||**), offering stronger theoretical guarantees and improved performance in practical scenarios.

**Questions:**

See the Weaknesses section above .

**Ethical Concerns:**

["NO or VERY MINOR ethics concerns only"]

**Final Justification:**

After reading the author's rebuttal and other reviewers' comments, I feel confident in recommending "Accept" for this paper. Accordingly, I raised my score from 4 to 5.

**Limitations:**

The authors appropriately address the limitations of their work.

**Paper Formatting Concerns:**

No significant formatting issues were noted.

**Quality:**

3

**Strengths And Weaknesses:**

**Strengths:**

1. The paper makes a significant contribution by providing the first proof of convergence for error feedback algorithms in distributed machine learning under generalized smoothness assumptions.

2. The theoretical analysis is robust, delivering strong convergence guarantees while avoiding restrictive assumptions such as data heterogeneity.

3. The paper is well-organized, with a clear structure. The theoretical contributions are supported by rigorous proofs, and the experiments provide valuable insights into the practical utility of the proposed algorithms.

**Weaknesses:**

1. While the paper includes experiments on several tasks, such as logistic regression and ResNet-20 on CIFAR-10, expanding the benchmarking to include a wider variety of problem domains would strengthen the generalizability of the results.

2. The paper touches on several important topics, including error feedback, generalized smoothness, communication compression, and normalization techniques. The authors could enrich the related work section by including more recent studies. For example, [1] uses the sign operation to compress communication and also provides guarantees under generalized smoothness assumptions, which should be discussed in the related work.

[1] Efficient Sign-Based Optimization: Accelerating Convergence via Variance Reduction. NeurIPS 2024.

---

> ### Author Rebuttal · Authors · 2025-07-30
>
> We sincerely thank the reviewer for the positive and encouraging feedback. We are glad that the reviewer found our work theoretically sound, well-organized, and of practical relevance.
>
>
> > **Q1**: While the paper includes experiments on several tasks, such as logistic regression and ResNet-20 on CIFAR-10, expanding the benchmarking to include a wider variety of problem domains would strengthen the generalizability of the results.
>
> **A1**: In the current manuscript, we evaluate our algorithms on three representative tasks—minimization of nonconvex polynomial functions, logistic regression with nonconvex regularization, and training ResNet-20 on CIFAR-10. These benchmarks were selected to demonstrate the effectiveness of normalization in stabilizing distributed error feedback under generalized smoothness, across both synthetic and real-world scenarios.
>
> While our primary focus is theoretical, we fully agree that broadening the empirical evaluation would further support the generality and practical relevance of our results.
>
>
>
>
> > **Q2**: The paper touches on several important topics, including error feedback, generalized smoothness, communication compression, and normalization techniques. The authors could enrich the related work section by including more recent studies. For example, [1] uses the sign operation to compress communication and also provides guarantees under generalized smoothness assumptions, which should be discussed in the related work.
>
> **A2**: We are happy to include additional recent studies that address generalized smoothness and normalization techniques. We will include and discuss [1] in the related work section for two main reasons.
>
> **Connection via  Normalization:** The sign operation can be viewed as coordinate-wise normalization. Potentially, our convergence results for normalized EF21 can be extended to the case where a sign operator is applied to the aggregated gradient descent update. The key difference lies in the trade-off: while the sign operator leads to a more aggressive compression scheme, it introduces a **dimension-dependent convergence rate**, which becomes a significant limitation in high-dimensional settings.
>
> **Parameter-Agnostic Guarantees:** Both [1] and our work provide parameter-agnostic convergence guarantees. That is, the algorithms do not require knowledge of the smoothness constants of the objective functions. This makes the methods more practical in distributed settings.

---

> > ### Comment · Reviewer_Acin · 2025-08-01
> >
> > After reading the author's rebuttal and other reviewers' comments, I feel confident in recommending "Accept" for this paper. Accordingly, I raised my score from 4 to 5.

---

> > > ### Author Response · Authors · 2025-08-04
> > >
> > > > After reading the author's rebuttal and other reviewers' comments, I feel confident in recommending "Accept" for this paper. Accordingly, I raised my score from 4 to 5.
> > >
> > >
> > > We are glad that our clarifications addressed your concerns. Thank you for recognizing our contributions and for raising your score from 4 to 5, thereby recommending the acceptance of our paper.

---

### Official Review · Reviewer_Sp22 · 2025-07-01

**Clarity:** 4
**Significance:** 4
**Originality:** 4
**Rating:** 5
**Confidence:** 2

**Summary:**

**Contributions**:

 1. Extends the results of Compressed Distributed Learning with Error Feedback (EF21)[1] to generalized smoothness, via normalization.
2. The proposed algorithm does not require additional assumptions like bounded gradient, bounded heterogeneity.
3. The proposed algorithm matches the convergence rate of EF21 with problem independent parameters (smoothness) which are generally unknown.
4. They also propose a stochastic variant (||EF21SGDM||) under the bounded expected gradient variance assumption with matching rates as the previous work[2]

[1] Ilyas Fatkhullin, Alexander Tyurin, and Peter Richtárik. Momentum provably improves error
333 feedback!
[2] |Richtárik, Peter, Igor Sokolov, and Ilyas Fatkhullin. "EF21: A new, simpler, theoretically better, and practically faster error feedback." _Advances in Neural Information Processing Systems_ 34 (2021): 4384-4396.|

**Questions:**

See weaknesses

**Ethical Concerns:**

["NO or VERY MINOR ethics concerns only"]

**Final Justification:**

All my concerns were answered by the reviewers.

**Limitations:**

Yes

**Quality:**

4

**Strengths And Weaknesses:**

**Strengths**:
1. The paper is clearly written, with improvements with prior work clearly mentioned.
2. Generalized smoothness is a much more relaxed condition and most FL works perform normalization/clipping and error feedback so this is an important analysis.
3. Numerical experiments show the strength of the proposed algorithm over EF21.

**Weaknesses/Questions**:
1. The authors provide single-step Federated Learning / Distributed Learning algorithm. However, clients perform multiple local steps before pushing updates to the server. Can this analysis be extended to multiple local steps without the bounded heterogeneity assumption?
2. This is not a weakness per se but a question, can the results be extended to strongly convex / convex losses / PL losses without a bounded gradient assumption  and achieve exponential convergence to the true minimum?

---

> ### Author Rebuttal · Authors · 2025-07-30
>
> We sincerely thank the reviewer for the positive and constructive feedback, and for recognizing the clarity of our presentation, the relevance of generalized smoothness in practical FL scenarios, and the empirical strength of our algorithms.
>
>
> > **Q1**: The authors provide a single-step Federated Learning / Distributed Learning algorithm. However, clients perform multiple local steps before pushing updates to the server. Can this analysis be extended to multiple local steps without the bounded heterogeneity assumption?
>
> **A1**: Thank you for this insightful question. We believe that our analysis can indeed be extended to incorporate multiple local steps **without requiring the bounded heterogeneity assumption**.
>
> To illustrate this, consider the following modification of the normalized EF21 algorithm:
>
> 1. Each client performs $T$ local gradient descent (local GD) steps using a fixed local stepsize $\eta$ after receiving the global model $x^k$ at iteration $k$.
>
>
> 2. Instead of communicating $\mathcal{C}^k(\nabla f_i(x^k) - g_i^{k-1})$, each client now transmits  $\mathcal{C}^k(  \frac{x^k - x_i^{k,T}}{\eta} - g_i^{k-1})$, where $x_i^{k,T}$ denotes the local model after $T$ steps of local GD starting from $x^k$.
>
> The primary change in the analysis occurs in **Lemma 5**, which bounds $\mathbb{E}\|\| \nabla f_i(x^{k+1})-g_i^{k+1} \|\|$.
>
> By the definition of $g_i^{k+1}$, by the triangle inequality, and then by the tower property of the expectation, we derive:
>
> $$
> \mathbb{E}\|\| \nabla f_i(x^{k+1}) - g_i^{k+1} \|\| \leq \mathbb{E}\|\| \nabla f_i(x^{k+1}) - \nabla f_i(x^k) \|\| + \mathbb{E}\|\| \nabla f_i(x^k) - \frac{x^k - x_i^{k,T}}{\eta}  \|\| + \mathbb{E}\left[ \sqrt{\mathbb{E}_k \|\| \frac{x^k - x_i^{k,T}}{\eta} - g_i^k - \mathcal{C}^k( \frac{x^k - x_i^{k,T}}{\eta} - g_i^k ) \|\|^2 }\right],
> $$
> where $\mathbb{E}_k[\cdot] = \mathbb{E}[\cdot| \mathcal{F}_k]$ and $\mathcal{F}_k$ is the filtration up to $k$.
>
> Next, by invoking the contractive property of compression, this simplifies to:
> $$
> \mathbb{E}\|\| \nabla f_i(x^{k+1}) - g_i^{k+1} \|\| \leq \mathbb{E}\|\| \nabla f_i(x^{k+1}) - \nabla f_i(x^k) \|\| + (1+\sqrt{1-\alpha})\mathbb{E}\|\| \nabla f_i(x^k) - \frac{x^k - x_i^{k,T}}{\eta}  \|\|.
> $$
>
> Both terms of the right-hand side can be upper-bounded:
>
>
> The first term can be bounded by the relaxed smoothness and the update rule for $x^{k+1}$.
> The second term can be bounded by leveraging analyses of local GD without relying on bounded heterogeneity, as in [1] and [2].
>
>
> Furthermore, it is a promising direction to integrate EF21 with other SOTA federated learning algorithms that accommodate multiple local steps and variance control, such as SCAFFOLD and ProxSkip.
>
> **References:**
>
> [1] Khaled, Ahmed, Konstantin Mishchenko, and Peter Richtárik. "First analysis of local GD on heterogeneous data." arXiv preprint arXiv:1909.04715 (2019).
>
> [2] Demidovich, Yury, et al. "Methods with local steps and random reshuffling for generally smooth non-convex federated optimization." arXiv preprint arXiv:2412.02781 (2024).
>
>
> > **Q2:** This is not a weakness per se but a question, can the results be extended to strongly convex / convex losses / PL losses without a bounded gradient assumption and achieve exponential convergence to the true minimum?
>
> **A2**: Thank you for raising this excellent point. Let us address the **convex** case and the **strongly convex / PL** cases separately.
>
> **1) Convex case:**
>
> Yes, our current analysis can be extended to convex objectives without assuming bounded gradients. Specifically, by the convexity of $f(\cdot)$, by Cauchy-Schwarz inequality, and by assuming that there exists a sequence $\{x^k\}$ such that $\|\| x^k-x^\star \|\| \leq R$ for some $R>0$, we obtain
>
> $$
> f(x^k)-f(x^\star) \leq \|\| \nabla f(x^k) \|\| \cdot \|\| x^k-x^\star\|\|.
> $$
>
> This inequality allows us to derive the convergence bound for $\min_{k}\mathbb{E}{f(x^k)-f(x^\star)}$ by applying it to Theorem 1 for || EF21 || and Theorem 2 for || EF21-SGDM ||. This approach enables sublinear convergence for minimizing convex, relaxed smooth functions.
>
> **2) Strongly Convex / PL case:**
>
> Extending our results to $\mu$-strongly convex or $\mu$-PL objectives is more subtle. While these settings satisfy:
> $$
> \|\| \nabla f(x^k) \|\|^2 \geq 2\mu (f(x^k) - f(x^\star)),
> $$
> Plugging this inequality into the convergence bounds in Theorems 1 and 2 yields convergence in terms of $\min_{k} \mathbb{E}{\sqrt{f(x^k)-f(x^\star)}}$. This approach does not imply exponential convergence to the optimum. Addressing this gap requires new techniques, potentially involving tighter Lyapunov functions or refined descent inequalities tailored to PL-type objectives. We believe this is an interesting open question and a promising direction for future investigation.

---

> > ### Comment · Reviewer_Sp22 · 2025-08-02
> >
> > Thanks for the detailed response. I continue to feel this should be accepted as a paper to NeurIPS.

---

> > > ### Author Response · Authors · 2025-08-04
> > >
> > > > Thanks for the detailed response. I continue to feel this should be accepted as a paper to NeurIPS.
> > >
> > > Thank you for your appreciation of our contributions and for recommending the acceptance of our paper. We are glad that our clarifications addressed your concerns.

---

### Decision · Program_Chairs · 2025-09-17

**Decision:**

Accept (poster)

**Comment:**

**Summary:**
The current sutidy of error feedback algorithms requires smoothness assumption which is not necessarily satisfied by for instance deep neural networks. Relying on a more general $(L_0, L_1)$-smoothness, the paper proposes a normalization strategy to an existing algorithm **EF21**, and studies the convergence property of the new scheme. Extension to stochastic setting is also considered. The new scheme is supported by numerical experiments.

**Strength:**
 - The paper makes a significant contribution the analysis of error feedback algorithms under merely generalized smoothness. The proposed analysis has potential impact to other problems.
 - Solid theoretical analysis without strong assumptions.
 - Convincing numerical experiments.

**Weakness:**
 - The presentation of the paper can be further improved.

**Discussion summary:**
All reviews acknowledge the significance of the proposed study, and its potential broad impact. We also suggest the reviewers to revise the paper according to the author-reviewer discussion.